## PROCEEDINGS A

climatology, statistical physics, mathematical modelling

climate modelling, multistability, quasi-potential theory, non-equilibrium systems, data-driven methods, manifold learning

**Author for correspondence:**
Valerio Lucarini
e-mail: v.lucarini@reading.ac.uk

# Dynamical landscape and multistability of a climate model

Georgios Margazoglou[1,2], Tobias Grafke[3],
Alessandro Laio[4] and Valerio Lucarini[1,2]

[1]Department of Mathematics and Statistics, and [2]Centre for the Mathematics of Planet Earth, University of Reading, Reading, UK
[3]Mathematics Institute, University of Warwick, Coventry, UK
[4]International School for Advanced Studies (SISSA), Trieste, Italy

GM, 0000-0002-1374-9374; TG, 0000-0003-0839-676X;
AL, 0000-0001-9164-7907; VL, 0000-0001-9392-1471

We apply two independent data analysis methodologies to locate stable climate states in an intermediate complexity climate model and analyse their interplay. First, drawing from the theory of quasi-potentials, and viewing the state space as an energy landscape with valleys and mountain ridges, we infer the relative likelihood of the identified multistable climate states and investigate the most likely transition trajectories as well as the expected transition times between them. Second, harnessing techniques from data science, and specifically manifold learning, we characterize the data landscape of the simulation output to find climate states and basin boundaries within a fully agnostic and unsupervised framework. Both approaches show remarkable agreement, and reveal, apart from the well known warm and snowball earth states, a third intermediate stable state in one of the two versions of PLASIM, the climate model used in this study. The combination of our approaches allows to identify how the negative feedback of ocean heat transport and entropy production via the hydrological cycle drastically change the topography of the dynamical landscape of Earth's climate.

# 1. Introduction

The climate, an extremely high-dimensional complex system, is composed of five interacting subdomains: a gaseous atmosphere, a hydrosphere (water in liquid form), a lithosphere (upper solid layer), a cryosphere (water in solid form) and a biosphere (ecosystems and living organisms) [1]. The climate is driven by the inhomogeneous absorption of incoming solar radiation and can be treated as a highly non-trivial dynamical system that features spatio-temporal variability on a vast range of scales. The system is at an approximate non-equilibrium steady state due to the resulting interplay of forcings, dissipation, positive and negative feedbacks, instabilities and saturation mechanisms [2]. The presence of periodic as well as irregular fluctuations in the boundary conditions does not allow the climate to reach an exact steady state [3,4].

A straightforward attempt to mathematically formulate the dynamics of the climate system is by defining a set of partial differential equations (PDEs) that describe the budget of mass, momentum and energy. As this set of PDEs is impossible to solve analytically, they are usually simulated numerically. Depending on the number of resolved variables, this procedure is extremely challenging both from a technological and scientific point of view, and requires a diversified approach. Therefore, a hierarchy of climate models can be established [5–8]. At the lowest level of such a hierarchy one can find simple zero or one-dimensional Energy Budget Models (EBMs) that describe in a highly simplified manner the fluxes of energy inside the climate system and and at its boundaries [9–11], as well as low-dimensional models that represent fundamental processes of the large-scale oceanic [12–14] and atmospheric dynamics [15–17]. Next come the so-called intermediate complexity models, which provide a parsimonious yet Earth-like representations of the dynamics of climate, see e.g. [18–25]. Finally, modern state-of-the-art climate models, similar to the ones featured in the latest Intergovernmental Panel on Climate Change (IPCC) report [26] are based on applying a series of necessary truncations and approximations in such a set of PDEs [27]. In general, the impact of the neglected scales of motions on the explicitly resolved scales is approximated via suitably developed parametrizations, which include deterministic, stochastic and possibly non-Markovian components [28,29].

## (a) Global stability properties of the climate system

The current astronomical configuration of Earth supports the present-day Warm (W) climate, and a frozen one, termed Snowball (SB), which exhibits global glaciation, extremely low temperatures and limited climatic variability. Geological and paleomagnetic evidence suggests that during the Neoproterozoic era (in particular around 630 and 715 Ma), the Earth exhibited at least two major long-lasting global glaciation periods, thus entering twice into the snowball climate state [30,31]. Simple energy balance models are able to reproduce the associated multistability of the climate system [9–11], which is mainly affected by the so-called ice-albedo feedback. The importance of such a mechanism is confirmed by studies performed with higher complexity models [30,32–35], including fully coupled climate models [36].

If we now focus on the current climate or the climate of the recent past (thus within the W state), the Earth is well known to feature further elements of multistability associated with critical transitions among stable states. Examples of geographically localized phenomena affecting the climate system featuring such a behaviour—the so-called tipping elements [37]—include the dieback of the Amazon forest [38], the shut-down of the thermohaline circulation of the Atlantic ocean [39], the methane release resulting from the melting of the permafrost [40] and the collapse of the atmospheric circulation regime associated with the Indian monsoon [41]. A critical transition taking place for one climatic subsystem may trigger the tipping of another element: this is the phenomenon of the so-called cascading tipping points [42,43].

Transitions between metastable states might be facilitated by mechanisms like stochastic resonance [44], which has been recently reframed according to the formalism adopted here for treating non-equilibrium systems [45]. Indeed, stochastic resonance is thought to act in the climate

system at different spatial and temporal scales, ranging from ultralong [46–48], to intermediate [49–52], to short ones [53–55].

In this work, we explore the multistability of a climate model through methods borrowed from non-equilibrium statistical physics, dynamical systems, and data science, thus pushing forward the scientific programme presented in [4,56]. We then take inspiration from the Waddington's 'epigenetic landscape' metaphor in evolutionary biology [57–60]. The phase space of the climate model can be explored by adding suitably defined stochastic forcing. As a result, the competing metastable climatic states can be viewed as vast valleys of a quasi-potential landscape $\Phi$, separated by mountain ridges, corresponding to unstable climates [56,61]. The stochastic forcing allows for exploring the landscape and, in particular, makes it possible to observe transitions between the metastable states.

Unfortunately, the actual evolution of the climate system cannot be fully regarded as the idealized stochastic motion in a fixed non-equilibrium quasi-potential landscape described above because geological, biological, astronomical and astrophysical factors modulate the landscape on a vast range of time scales. Nonetheless, the quasi-potential landscape viewpoint can be extremely useful to understand its multistability at an instance in time.

## (b) Outline of the paper

In this paper, we will study the transitions between competing metastable states of PLASIM [25], a simplified climate model that has shown extreme flexibility in describing the dynamics of a vast range of climate conditions, including very exotic ones [62–67]. The model features $O(10^5)$ degrees of freedom (d.f.). We consider two set-ups of the model—one allowing for the ocean to transport heat from low to high latitudes (set-up A), previously used in [66], and one where only the atmosphere is able to perform large-scale heat transport (set-up B), previously used in [65]. The main limitation of the model is its lack of explicit representation of the deep ocean circulation, which is very relevant for climate on multidecadal to millennial time scales.

We explore the phase space of the model by allowing the solar irradiance $S^*$ to randomly fluctuate around the present-day mean value of $S^* = 1365\,\text{W/m}^2$, thus triggering transitions among the competing climate states. Following [56,61], we construct the quasi-potential of the stochastically perturbed system [68–71], and we compute by stochastic averaging the mean transitions paths among attractors, which are composed of instantonic and relaxation trajectories.

The identification of the competing attractors is approached in two ways. First, we use standard forward numerical modelling and identify different asymptotic states, which are associated, when the dynamics is deterministic, with separate basins of attraction. Second, competing attractors are automatically detected through data-driven methods applied to the output of long stochastic integrations of the model. Such methods have been used for studying metastable states in biomolecules, and allow one to reconstruct very effectively the quasi-potential $\Phi$ of the system, partially taking care of the *curse of dimensionality* [72–75]. We anticipate that whereas in set-up A we find the two usual W and SB states, set-up B has a third stable climate state (to be termed 'cold climate' (C) in the following), with an ice-free latitudinal band at roughly $\pm 20°$ around the Equator featuring mild surface temperatures, a vigorous atmospheric circulation and non-trivial hydrological cycle. Such a third state resembles previously suggested exotic climatic configurations such as the slushball Earth [32] and the Jormungand state [34]. The C state corresponds to a shallow minimum of the quasi-potential and disappears when ocean transport is included in the system, which acts as a strong stabilizing mechanism. The presence of the C state has important implications both on the statistical mechanics of the system and on the topology of the transition paths between the W and the SB states.

The paper is structured as follows. Section 2 contains the mathematical framework behind our analysis. Section 3 provides a description of the climate model used in this study. Section 4 contains the description and critical analysis of the results. Section 5 is dedicated to drawing the conclusion of this work and to presenting future research perspectives. The electronic supplementary material attached to this paper contains extra information on the numerical

simulations, on the computation of the average transition paths, as well as a brief and informal description of the mathematics of the transfer operator and of its finite-size representation. Additionally, it includes links to a set of movies related to the numerical simulations performed in this study.

## 2. Qualitative and quantitative aspects of the multistability of the climate system

### (a) Dynamical landscape of the climate system

Let us consider a deterministic dynamical system defined by the following set of ordinary differential equations:

$$\frac{\mathrm{d}x}{\mathrm{d}t} = F(x, t), \quad x(t=0) \equiv x_0, \tag{2.1}$$

where $x(t) \in \mathbb{R}^N$ describes the state of the system at time $t$ with initial condition $x_0$, and $F(x, t) \in \mathbb{R}^N$ is a smooth vector field. The initial condition $x_0$ determines the asymptotic state of its orbit. We assume that the system is forced and dissipative, so that $N-$ volumes in phase space are contracted by the flow. If equation (2.1) possesses more than one asymptotic state, defined by the attractors $\Omega_j$, $j = 1, \ldots, J$, the system is multistable. The phase space is partitioned between the basins of attraction $B_j$ of the attractors $\Omega_j$ and the boundaries $\partial B_l$, $l = 1, \ldots, L$ separating such basins, which possess a set of saddle points $\Pi_l$, $l = 1, \ldots, L$. Such saddle points attract initial conditions on the basin boundaries [76–78] and can be computed using the so-called edge tracking algorithm [79], which was used in an EBM by Bódai *et al.* [80]. Chaotic unstable saddles, then termed Melancholia (M) states, have been constructed with the edge tracking algorithm for a simplified climate model built by coupling a primitive equation atmosphere with a diffusive ocean [35].

Escaping an attractor is possible if the system undergoes a properly defined stochastic forcing [81–83] . By subjecting equation (2.1) to a Gaussian random noise and considering it in Itô form, we write the stochastic differential equation

$$\mathrm{d}x = F(x)\,\mathrm{d}t + \sigma s(x)\,\mathrm{d}W, \tag{2.2}$$

where $\mathrm{d}W$ is the increment of an $M$-dimensional Wiener process, $F(x)$ is in this context usually referred to as the drift term, $C(x) = s(x)s(x)^{\mathrm{T}} \in \mathbb{R}^{N \times N}$ is the noise covariance matrix where in general the volatility matrix $s(x) \in \mathbb{R}^{N \times M}$, and $|\sigma| > 0$ determines the strength of the noise.

In the present work, we introduce stochasticity in the form of a fluctuating solar constant, which amounts to considering only one independent Brownian motion, so that $s(x) \in \mathbb{R}^{N \times 1}$ and $C(x)$ is rank one. Additionally, only the d.f. directly associated with the incoming solar radiation are directly impacted by the stochastic forcing. As clarified in [56], we expect that the applied noise percolates to all d.f.'s of the system as a result of non-degenerate interplay between stochastic forcing and the deterministic component of the dynamics given by the drift term, so that we can assume that we are dealing with a hypoelliptic diffusion process [84]. Hence, we expect that for $|\sigma| > 0$ the invariant measure of the system is smooth.

We now follow [69–71,85], consider the weak-noise limit, and express the stationary solution of the Fokker–Planck equation [86] corresponding to equation (2.2) as a large deviation law

$$\rho_\sigma(x) \sim Z(x) \exp\left(-\frac{2\Phi(x)}{\sigma^2}\right), \tag{2.3}$$

where $Z(x)$ is a pre-exponential factor and $\Phi(x)$ is the quasi-potential, a non-equilibrium generalization of the notion of free energy. $\Phi(x)$ can be obtained as a nontrivial solution of the the the Hamilton–Jacobi equation [70,87] $F_i(x)\partial_i\Phi(x) + C_{ij}(x)\partial_i\Phi(x)\partial_j\Phi(x) = 0$. See [68,85] for a detailed

discussion on the regularity of $\Phi$, and [88] for an alternative approach based on variational arguments. It is possible to write the drift vector field as the sum of two vector fields:

$$F_i(\mathbf{x}) = R_i(\mathbf{x}) - C_{ij}(\mathbf{x})\partial_j \Phi(\mathbf{x}), \quad R_i(\mathbf{x})\partial_i \Phi(\mathbf{x}) = 0. \tag{2.4}$$

A different strategy for attaining the decomposition of the drift term into a symmetric and an antisymmetric component has been proposed by Ao [89] and Yuan *et al.* [90].

In the case one switches off the noise, $\Phi$ acts as a Lyapunov function whose decrease with time describes the convergence of an orbit to an attractor. Indeed, $\Phi$ has local minima at the deterministic attractors $\Omega_j, j = 1, \ldots, J$, and has a saddle behaviour at the saddles $\Pi_l, l = 1, \ldots, L$. If an attractor or a saddle is chaotic, $\Phi$ has constant value over its support, which can then be a strange set [69,85].

A special class of trajectories, named instantons, define, in the zero-noise limit, the most probable way to exit an attractor [82,91]. An instanton connects an attractor $\Omega$ to a point $\mathbf{x}$ within the same basin of attraction and can be obtained by minimizing the action of the stochastic field theory associated with the system [88,92–94]. The instantonic trajectory obeys the equation of motion $\mathrm{d}x_i/\mathrm{d}t = R_i(\mathbf{x}) + C_{ij}(\mathbf{x})\partial_j \Phi_\Omega(\mathbf{x})$, which has a reversed component of the gradient contribution with respect to the drift field, see equation (2.4). If $R(\mathbf{x})$ vanishes, instantonic trajectories follow the same path (in reverse direction) with respect to relaxation trajectories, which is a basic characterization of equilibrium systems and detailed balance.

Within the basin of attraction of $\Omega$ one can define the local quasi-potential $\Phi_\Omega(\mathbf{x})$ as the action for the instanton linking $\Omega$ and $\mathbf{x}$ [88]. Escapes from an attractor $\Omega$ occur via the saddle $\Pi$ situated at the corresponding basin boundary having the lowest value of the barrier height $\Delta\Phi_{\Omega\to\Pi} = \Phi_\Omega(\Pi) - \Phi_\Omega(\Omega)$ [78] and are Poisson-distributed events, where the probability that an orbit does not transition up to time $t$ is, similarly to the classic Kramers' Law [95], is given by

$$P(t) = \frac{1}{\bar{\tau}_\sigma} \exp\left(-\frac{t}{\bar{\tau}_\sigma}\right), \quad \text{with } \bar{\tau}_\sigma \propto \exp\left(\frac{2\Delta\Phi_{\Omega\to\Pi}}{\sigma^2}\right). \tag{2.5}$$

Unfortunately, in the case of multistable systems, one cannot, in general, simply read off the barrier height $\Delta\Phi_{\Omega\to\Pi}$ from the $\Phi(\mathbf{x})$ of equation (2.3), because glueing together the various local quasi-potentials does not give the global quasi-potential $\Phi(\mathbf{x})$ [71,85]. The local and global notions of quasi-potential can be brought to a common ground if the system is at equilibrium so that no global probability fluxes are present. Equivalence between the information provided by the local and global quasi-potentials is also realized if the system is not an equilibrium one but only two competing states are present with a single saddle embedded in the boundary between the two basins of attraction, as in the cases analysed in [56,61]. In general, we will resort to measuring separately the invariant measure (2.3) and the barrier heights (2.5).

## (b) Exploring the topography of the quasi-potential

To study the topography of $\Phi$, one can neglect the pre-exponential factor $Z(\mathbf{x})$ in equation (2.3) and project the invariant measure $\rho_\sigma(\mathbf{x})$ on a—possibly small—number $n$ of pre-selected variables defined by the function $s = S(\mathbf{x}) \in \mathbb{R}^n$. This gives

$$\Phi(s) \sim -\frac{\sigma^2}{2} \log \rho_\sigma(s) = -\frac{\sigma^2}{2} \log \int \mathrm{d}x \delta(S(\mathbf{x}) - s)\rho_\sigma(\mathbf{x}). \tag{2.6}$$

If $n$ is small, $\rho_\sigma(s)$ can be efficiently estimated, e.g. by computing a histogram. Its minima and saddle points can then be found straightforwardly, even by visual inspection. However, this approach has a key drawback: the choice of the variables used for the projection is arbitrary, and multiple attractors may appear erroneously merged for a too low-dimensional choice, see below.

To circumvent this problem, one can follow an approach borrowed from manifold learning, which allows estimating the quasi-potential as a function of a large number of variables and studying its topography directly in such a space. As shown below, this allows identifying the deterministic attractors of a system of the form given in equation (2.2) without preselecting a small number of putative important variables, i.e. it is applicable even when $n \gg 1$.

This procedure is rooted on a general property of dynamical systems. Even if the dynamics takes place in an $N$-dimensional space, where $N$ can be very large, the trajectory is often contained in an embedding manifold of dimension $d$ where typically $d \ll N$ [96]; in the case of deterministic chaos, this information in encoded by the Kaplan–Yorke dimension [97]. This makes the estimate of $\rho_\sigma$ *restricted to the manifold* numerically and algorithmically possible. However, this manifold is typically twisted and curved, and it is very difficult (or even impossible, if the topology of the manifold is non-trivial) to define a global coordinate chart. We can, instead, estimate $\Phi$ directly on the embedding manifold as in equation (2.6) without defining explicitly the function $S(x)$.

Consider a trajectory $\mathbf{x}_t$, where $t$ labels the different configurations. Consider the Euclidean distance $r_{t,t'} = \|\mathbf{x}_t - \mathbf{x}_{t'}\|$ between pairs of configurations. Even if this distance is defined in a $N$-dimensional space, if $\mathbf{x}_t$ and $\mathbf{x}_{t'}$ are so close that one can neglect the curvature, $r_{t,t'}$ approximates a metric on the manifold. Building on this approximation, one first estimates $d$ from the statistics of the ratio between the distance of the nearest neighbour $r_{t,(1)}$ of each data point $t$ and the distance of its second nearest neighbour $r_{t,(2)}$. One can prove that $\mu_t = r_{t,(2)}/r_{t,(1)}$ is Pareto distributed [72]: $\mu_t \sim \mathrm{PD}(d)$, except for a correction which depends on the curvature of the manifold and on the variation of the invariant measure on the scale of distance $r_{t,(2)}$. These errors vanish in the limit of infinite sample size [72]. This allows inferring the value of $d$ from the empirical probability distribution of $\mu$; see closely related results in [98,99].

The next step is estimating the quasi-potential $\Phi_t \sim -(\sigma^2/2)\log(\rho_\sigma(x_t))$. This is done using the approach in [73], a generalization of the $k$-nearest neighbour density estimator [100] in which the probability density is estimated implicitly on the embedding manifold and the optimal $k$ becomes configuration-dependent. The optimal $k$ is defined by finding, via a statistical test, the largest neighbourhood of $x_t$ in which the density can be considered constant with a given statistical confidence. We denote by $\mathcal{N}_t$ this neighbourhood and by $\hat{k}_t$ the optimal value of $k$ for configuration $t$. $\Phi_t$ is then obtained by maximizing a likelihood with respect to two variational parameters [73]:

$$\Phi_t = \operatorname*{argmax}_\phi \max_a \left( -\phi\hat{k}_t + a\frac{\hat{k}_t(\hat{k}_t + 1)}{2} - \sum_{l=1}^{\hat{k}_t} e^{-\phi + al} v_{t,l} \right) \tag{2.7}$$

where, denoting by $\Omega_d$ the volume of a $d$-sphere of unitary radius and by $r_{t,(l)}$ the distance between $x_t$ and its $l$th nearest neighbour, $v_{t,l} = \Omega_d(r_{t,(l)}^d - r_{t,(l-1)}^d)$. Notice that if one takes $a = 0$ equation (2.7) gives $\Phi_t = -\log(\hat{k}_t/V^{\hat{k}_t})$, where $V^{\hat{k}_t} = \sum_{l=1}^{\hat{k}_t} v_{t,l}$ is the volume enclosed in a $d$-sphere of radius equal to the distance between the configuration $t$ and its $\hat{k}_t$th neighbour. Therefore, in these conditions, the quasi-potential is estimated as minus the logarithm of the density estimated by a standard $k$-NN estimator. The $a$-dependent term allows taking into account linear variations of the density in the neighbourhood. Importantly, this procedure provides, within the same statistical framework used for defining the likelihood in equation (2.7), an estimate of the error on $\Phi_t$, which we denote by $\varepsilon_t$.

The final step is inferring the topography of the quasi-potential from the estimates $\Phi_t$. This is done through an unsupervised extension of density peak clustering [74,75]. Configuration $t$ is assumed to be a local minimum of $\Phi$ if the following two properties hold: (I) $\Phi_t < \Phi_{t'} \ \forall x_{t'} \in \mathcal{N}_t$, namely if $x_t$ is a minimum in $\mathcal{N}_t$, (II) $x_t \notin \mathcal{N}_{t'} \ \forall t' : \Phi_{t'} < \Phi_t$, namely if $x_t$ does not belong to the $\mathcal{N}_{t'}$ neighbourhood of any configuration with lower $\Phi$. An integer label $c$ is assigned to each of the $n$ local minima found in this manner. The labels of the other configurations are found iteratively, by assigning to each point the same label of its nearest neighbour of smaller $\Phi$ [75]. The set of points with the same label $c$ is denoted by $\mathcal{A}_c$ and is assumed to correspond to a basin of attraction. The saddle points between the attractors are then found. A configuration $x_t \in \mathcal{A}_c$ is assumed to belong to the border with a different attractor $\mathcal{A}_{c'}$ if there exists a configuration $x_{t'} \in \mathcal{N}_t \cap \mathcal{A}_{c'}$ such that $r_{t,t'} = \min_{x_{t''} \in \mathcal{A}_c} r_{t'',t'}$. The saddle point between $\mathcal{A}_c$ and $\mathcal{A}_{c'}$ is the point of minimum $\Phi$ belonging to the border between the two attractors.

Finally, the statistical reliability of the attractors is assessed as follows. Denote by $\Phi_c$ the minimum value of $\Phi_t$ in the attractor $c$, by $\varepsilon_c$ its error, by $\Phi_{c,c'}$ the value of $\Phi_t$ of the saddle point

between $\mathcal{A}_c$ and $\mathcal{A}_{c'}$ and by $\varepsilon_{c,c'}$ its error. If $\Phi_{c,c'} - \Phi_{c'} < Z\sqrt{\varepsilon_c^2 + \varepsilon_{c,c'}^2}$, the attractor $c'$ is merged with attractor $c$ since the value of the quasi-potential at its minimum and at the saddle point are indistinguishable at a statistical confidence defined by $Z$ [74]. The process is repeated until all the attractors satisfy this criterion, and are therefore statistically robust with a confidence $Z$.

The whole procedure enables us to detect metastable states that might be masked in a low-dimensional projection of the invariant measure. In the case the analysed data have been produced using a numerical model (as is the case here), it is possible to have conclusive results on the correctness of a candidate attractor by running noiseless forward simulations from the best estimate of its position (and nearby points) and observe whether it indeed persists indefinitely.

## 3. The climate model

We perform the numerical simulations using PLASIM, an open-source intermediate complexity climate model developed at the University of Hamburg [25]. PLASIM has a total of $\mathcal{O}(10^5)$ d.f., and retains some of the most important features of the climate, but is considerably less sophisticated and cheaper to run than the present state-of-the-art Earth System Models that reach more than $\mathcal{O}(10^8)$ d.f. [27]. PLASIM is extremely flexible and has been used for studying a rather wide range of climatic conditions [62–67], hence providing the perfect testing ground for novel theoretical investigations in climate science. PLASIM is well known to feature multistable dynamics, which has been thoroughly discussed in previous studies [8,33,63].

The dynamical core of PLASIM is responsible for describing the mass and the budgets of momentum, energy and water in the atmosphere. The primitive equations are solved by the spectral transform method [101] in the horizontal, by finite differences in the vertical and for the time advancing scheme, a semi-implicit time stepping is used [102]. Further to that, unresolved physical processes, e.g. horizontal and vertical diffusion, long- and short-wave radiation, interaction with clouds, moist processes and dry convection, precipitation, boundary layer fluxes of latent and sensible heat, and a land surface with biosphere are among the many to be effectively parametrized into the model. In that way, PLASIM simulates with a fair degree of accuracy all the necessary components of a realistic Earth-like climate system, with the notable exception of a dynamical component able to simulate the deep oceanic circulation; see discussion below. As it will become apparent below, the presence in PLASIM of a reasonably realistic representation of the hydrological cycle is key to introducing a new layer of complexity in the present study compared to what had been explored in previous investigations of the global stability properties of the climate systems [35,56,61].

Our experimental configuration uses a present-day geography and consists of a 50 m deep one-layer slab ocean model, which includes a thermodynamic sea ice module [103]. The resolution of the model is T21 in the horizontal direction, corresponding to a $5.6° \times 5.6°$ grid cell, with 10 atmospheric levels in the vertical, while the time-step is 45 min. Finally, we fix the $CO_2$ concentration to 360 ppm, while daily and seasonal cycles have been purposefully neglected to further remove any explicit time dependency of the evolution equations.

We configure two experimental set-ups that differ in terms of the oceanic heat transport. In set-up A the horizontal ocean diffusion is active and its parametrization requires choosing a specific value for the horizontal diffusivity constant. This set-up allows for a simple yet effective representation of the impact of the large-scale ocean transport on the climate as a whole, and has been used in a recent study where response theory was used to perform climate projections [66]. In set-up B, the horizontal ocean diffusivity is set to 0, which implies that the negative feedback associated with the large-scale oceanic heat transport is switched off. A similar configuration as in set-up B has been previously employed to study the thermodynamic properties of the climate in response to controlled changes of the solar constant [33] or of the $CO_2$ concentration [62,65].

Following [56,61], the stochastic forcing needed to explore the phase space of the system is introduced as random fluctuations of the solar irradiance around its present-day value $S_0^* = 1365\,\mathrm{W/m^2}$. Each year, a different value is prescribed according to $S^* = S_0^* + \eta$ , where $\eta$ is a

random number drawn from a normal distribution with vanishing mean and standard deviation $\delta S = \sigma S_0^*$. We consider a vast range of values for $\sigma$, ranging from 0.01 to 0.26, and perform multiple simulations with duration ranging from hundreds to tens of thousands of years, in order to explore the local and the global properties of the phase space of the system. Note that when weaker noise is considered, the exploration of the phase space requires longer simulations, as the transitions between the basins of attraction become exponentially rarer, see discussion below.

## 4. Results

## (a) Set-up A: atmospheric and oceanic large-scale energy transport

### (i) The two competing climate states

In set-up A, the representation of the large-scale oceanic energy transport is, euphemistically, oversimplified compared to what really occurs in Earth, as our model cannot represent the process of deep water formation and the large-scale ocean circulation [104–106]. Nonetheless, the presence of horizontal heat diffusion performed by the ocean has the merit of introducing an additional mechanism—on top of atmospheric transport fuelled by baroclinic instability—that contributes to reducing the large-scale temperature difference between low and high latitudes [1,107–109]. Using a large set of initial conditions ranging from very cold to very warm and taken from the restart files of the simulation contained in [33], we found empirical evidence of (only) two competing asymptotic states corresponding to the W and SB climates, in agreement with a plethora of previous investigations, see §1. The lack of a realistic dynamic ocean implies that PLASIM misses the multidecadal, ultra-low frequency relaxation and oscillatory modes associated with the advective feedbacks of the ocean near the W climate, see discussion in [110]. Additionally, as discussed in §5, the presence of a dynamic ocean might lead to additional features in the quasi-potential near the W climate, associated with tipping points. Instead, one expects that the lack of a dynamic ocean is less critical near the SB climate, because no large-scale circulation is present when the surface of the Earth is entirely frozen.

In figure 1, we present the zonally averaged annual mean of a 40-year long time-series of several observables, computed when steady-state conditions are realized in the absence of stochastic forcing ($\sigma = 0$). We compare here zonally averaged fields of the W climate (red lines) and of the SB (blue lines); additional information on globally averaged quantities are presented in table 1. Figure 1a shows the climatology of the zonal mean surface temperature. In agreement with previous studies performed on PLASIM [8,33,63], the SB state features global glaciation and extremely low temperatures at all latitudes, while the W state is similar to the present-day climate; see also the map of sea ice cover in figure 2, where the limit of sea ice approximately coincides with the isoline of 0°C in the surface temperature shown in figure 1a.

Figure 1b shows the annual mean budget of the precipitation minus evaporation rate (P–E) as well as the annual zonally averaged precipitation. The SB climate is almost entirely dry, because a very cold atmosphere can retain only a tiny amount of water vapour, as a result of the Clausius–Clapeyron relation [1]. The W climate has the familiar maximum of precipitation in the equatorial belt and secondary peaks in the mid-latitudes, resulting from convective precipitation and synoptic disturbances, respectively. The P-E field describes the scenario of net water vapour transport from the tropics into the equatorial belt and into the mid-latitudes [1].

Figure 1c shows the zonally averaged net energy budget at the top of the atmosphere (TOA), which is the sum of the incoming shortwave radiation and the outcoming longwave radiation and scattered shortwave radiation. Note that the fluxes are positive (negative) when entering (leaving) the planet. At steady state, the zonal TOA energy imbalance is compensated by the divergence of the meridional atmospheric enthalpy transport [1,107,108]. Such a transport is much stronger for the W state, where large contributions come from baroclinic eddies and from the large-scale transport of water vapour. Baroclinic eddies are located in the region of the jet, where zonal winds in the upper troposphere at 300 hPa (near the tropopause, where

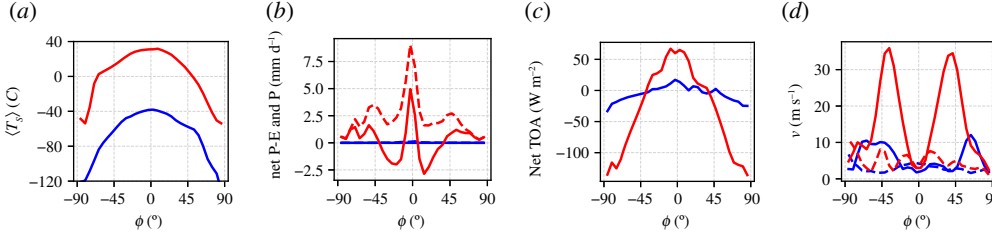

**Figure 1.** Climatology of the zonal averages of (*a*) surface temperature, (*b*) P-E (solid lines) and Precipitation (P, dashed lines), (*c*) TOA net radiation, (*d*) magnitude of zonal wind speed at 300 hPa (solid lines) at 1000 hPa (near surface, dashed lines) versus the latitude $\phi$. Blue lines: SB state. Red lines: W state. (Online version in colour.)

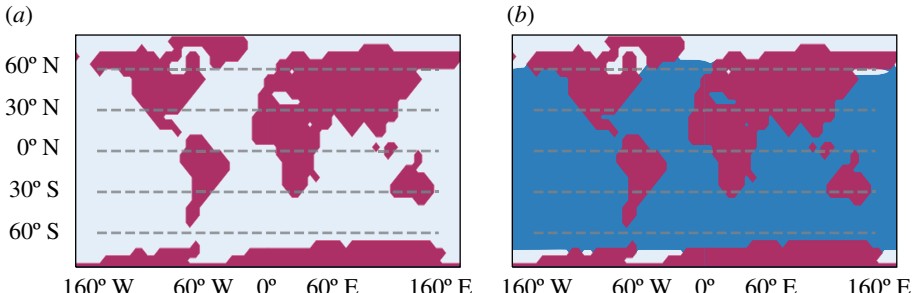

**Figure 2.** Sea ice coverage comparison between (*a*) snowball and (*b*) warm climates, where the colour coding is white for ice, blue for sea and red for land. We depict the land-map used by our model. (Online version in colour.)

**Table 1.** Main climatic features of the stable climates for the two experimental configurations in absence of stochastic forcing ($\sigma = 0$), where A refers to set-up A, and B to set-up B; W for warm state, C for cold state and SB for snowball state; LEC stands for Lorenz energy cycle.

|  | $[\langle T_S \rangle]$ (°C) | $\Delta T_{EP}$ (°C) | sea ice (%) | LEC (W m$^{-2}$) |
|---|---|---|---|---|
| A  W | 15.0(2) | 26.4(3) | 5.5(1) | 3.39 |
| A  SB | −55.2(3) | 25.7(5) | 100 | 1.00 |
| B  W | 4.4(3) | 40.0(5) | 27.7(1) | 4.79 |
| B  C | −28(2) | 53(1) | 70(2) | 3.79 |
| B  SB | −52.5(5) | 25.9(5) | 100 | 1.19 |

the peak intensity is found)—figure 1*d*—and their existence is made possible by the conversion of available potential into kinetic energy via baroclinic instability, which is associated with the presence of a substantial meridional temperature difference between low and high latitudes. The vigorous circulation of the W state corresponds to a powerful Lorenz energy cycle [111] ($\approx 3.4$ W m$^{-2}$). Instead, the meridional enthalpy transport and the zonal circulation of the SB state are extremely weak, corresponding to very modest meridional temperature gradients [8,33,63]. The SB state features a very weak Lorenz energy cycle ($\approx 1.0$ W m$^{-2}$), as weak meridional temperature gradients lead to a scarce reservoir of available potential energy and shuts down the mechanism of baroclinic instability. Correspondingly, surface winds are much weaker in the SB than in W climate (figure 1*d*).

## (ii) Noise-induced transitions

In what follows, we will apply a very severe coarse-graining to the phase space of the model. Indeed, we perform a projection on the plane spanned by the globally and 30-day averaged

surface temperature $[\langle T_S \rangle]$ and 30-day averaged Equator minus Poles surface temperature difference $\Delta T_{EP} = \langle T_{Eq} \rangle - \langle T_{Po} \rangle$, where we denote the spatial average of the field $X$ by $[X]$, and the temporal average by $\langle X \rangle$. Specifically, $T_{Eq} = [T_S]_{30°S}^{30°N}$ and $T_{Po} \equiv ([T_S]_{30°N}^{90°S} + [T_S]_{90°S}^{30°S})/2$. Such a projection allows retaining a minimal yet still physically relevant description of the system [35,56,61,80]. Indeed, variations in the globally averaged surface temperature reflect, to a first approximation, changes in the energy budget of the planet (warming versus cooling), while $\Delta T_{EP}$ controls the large-scale energy transport performed by the geophysical fluids [1,108].

The asymptotic state of the system in absence of stochastic forcing corresponds to one of the attractors described above and is determined by the initial condition. Transitions between the attractors can be induced by noise. Figure 3a portrays the normalized projection of the invariant measure of the stochastically forced system ($\sigma = 18\%$) on the phase space spanned by $[\langle T_S \rangle]$ and $\Delta T_{EP}$, while figure 3b portrays the quasi-potential estimated using equations (2.3) and (2.6):

$$\Phi([\langle T_S \rangle], \Delta T_{EP}) \sim -\frac{\sigma^2}{2} \log \rho_\sigma ([\langle T_S \rangle], \Delta T_{EP}), \tag{4.1}$$

where the global minimum is set to 0. The noise level given by $\sigma = 18\%$ is the lowest allowing for a detailed global exploration of the phase space within a—for us—reasonably long $(O(3 \times 10^4\ y)$ simulation, as one observes a good number $(O(40))$ of transitions between the competing states. We find that the basin of the W attractor is deeper (lower values of the quasi-potential) compared to the basin of the SB attractor. By using equation (2.5) and performing an exponential fit of the average residence times in the two attractors for different values of the noise intensity, we obtain $\Delta \Phi_{W \to SB} \approx 700(40)$ and $\Delta \Phi_{SB \to W} \approx 240(50)$. The good quality of the fit confirms that the weak-noise approximation is valid; see figure 3c

It is then worth looking at the paths of the SB $\to$ W and W $\to$ SB transitions. In the weak-noise limit, the stochastic average of the escape trajectories gives the instantonic path for the portion of trajectory connecting the initial attractor to an M state, and the relaxation path for the remaining part of the trajectory, which connects the M state to the final attractor. The red (blue) line in figure 3b indicate the stochastic averages of the SB $\to$ W (W $\to$ SB) transition trajectories. The procedure for computing the average paths is described in detail in the electronic supplementary material.

As discussed above, escape trajectories and relaxation trajectories are expected to follow different paths in general non-equilibrium systems. We are indeed able to find such an essential feature of non-equilibrium systems, as clearly detailed in figure 3b. In simpler set-ups with a unique saddle, the crossing point between the red and the blue line must correspond to the position of the M state [56,61]. Here the crossing between the two transition paths as observed in figure 3b is an artefact of looking at that specific two-dimensional projection; the three-dimensional projection of the phase space in figure 3d instead reveals that the SB→W and the W→SB transition paths do not intersect because they go through two different M states. This is a major difference with respect to the analysis performed in [56,61]. We have a clear indication that in the model used here large-scale currents are present in the phase space, which characterize non-equilibrium conditions; see [112] for an application of this concept in a climatic context.

It is reasonable to ascribe such a difference to the fact that here we are able to include a large class of processes associated with the transport of water and with its phase changes between solid, liquid and gaseous forms. Indeed, the hydrological cycle is to a great extent responsible for the irreversibility of atmosphere [2,113,114] and, at more quantitative level, overwhelmingly contributes to the total entropy production of the geophysical fluids compared to the dissipation of kinetic energy and the turbulent exchange of sensible heat [62,63,115,116]. We argue that the lack of a comprehensive treatment of water in the model used in [56,61] leads to an underestimation of the actual entropy production of the system, which makes it closer to equilibrium than the model considered here. According to a statistical mechanics angle, one sees this as associated with the absence (or significant reduction) of probability currents, which are largely suppressed by the presence of a single saddle separating the competing basins of attraction.

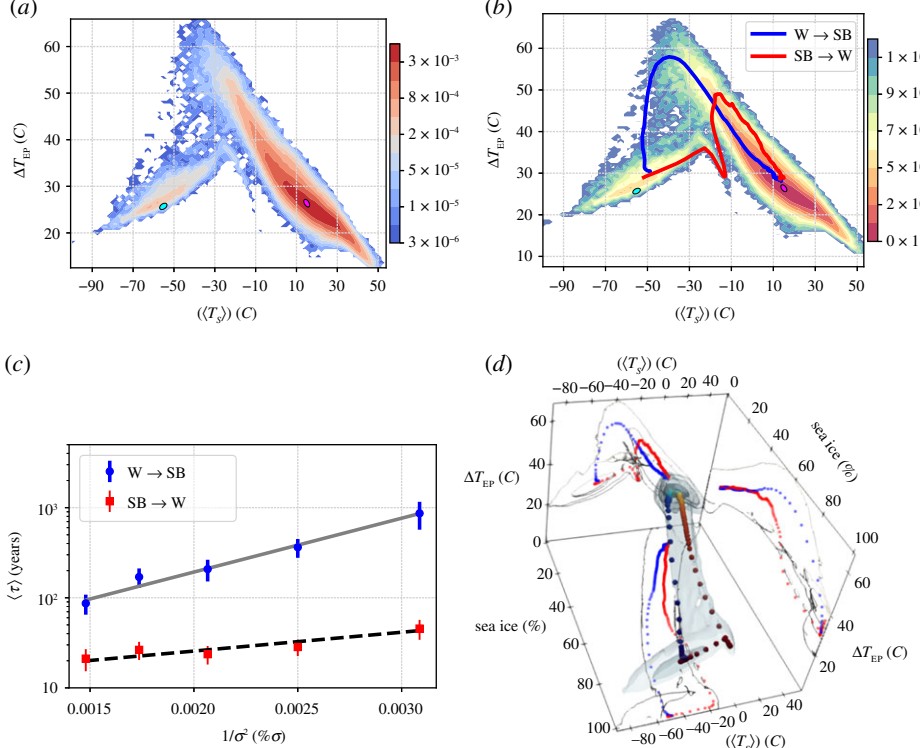

**Figure 3.** Set-up A. (*a*) Two-dimensional projection of the invariant measure on ([$\langle T_S \rangle$], $\Delta T_{EP}$); $\sigma = 18\%$. (*b*) Quasi-potential $\Phi$, whose global minimum is set to 0. The blue (red) line corresponds to stochastically averaged transition paths for the W→SB (SB→W) transitions. The coloured ellipses indicate the location of the deterministic attractors corresponding to SB state (cyan) and W state (magenta). (*c*) Average escape time versus the inverse squared %$\sigma$, where dashed black and straight grey lines correspond to fitting equation (2.5). (*d*) Transition paths SB → W (red) and W → SB (blue) in the three-dimensional space spanned by [$\langle T_S \rangle$], $\Delta T_{EP}$, and the arctic sea ice percentage for $\sigma = 18\%$. The shading indicates the density of the projected invariant measure, while a two-dimensional projection of the transition paths in each plane is added. (Online version in colour.)

The presence of clear distinction between the SB→W and the W→SB transition paths indicates that the global thawing and the global freezing of the planet are fundamentally different processes; see the movies that can are linked from the caption of figure S4 in the electronic supplementary material. The thawing proceeds as follows. First, because of persistent positive anomalies of the solar irradiance, the global temperature of the planet grows without much changes in $\Delta T_{EP}$, as the atmospheric circulation is extremely weak and the oceanic transport is absent. Then, the equatorial belt starts to melt and, due to the local large decrease of the albedo and subsequent intense warming, $\Delta T_{EP}$ increases substantially—see the almost vertical portion of the red line in figure 3b. This leads to a strong enhancement of the meridional heat transport performed by the atmosphere and by the ocean, which causes the thawing of the sea ice at higher latitudes until the sea ice line reaches very high latitudes compatible with the W climate.

The global freezing of the planet, instead, proceeds in the following way. The cause of the freezing is, obviously, the presence of a (rare) persistent negative anomaly of the solar irradiance. The reduction of incoming solar radiation has an amplified effect at high latitudes, because of the ice-albedo feedback, leading to an increase of $\Delta T_{EP}$. The increase in $\Delta T_{EP}$ causes a strengthening in the meridional heat transport, which acts as a stabilizing feedback—see the diagonal portion of the blue line in figure 3b. Nonetheless, if the anomaly in the solar irradiance is sufficiently strong and persistent, the sea ice line moves equatorward, until the equatorial belt freezes and undergoes further extreme cooling because the albedo becomes very high, leading eventually to a very low value of $\Delta T_{EP}$ in the final SB state.

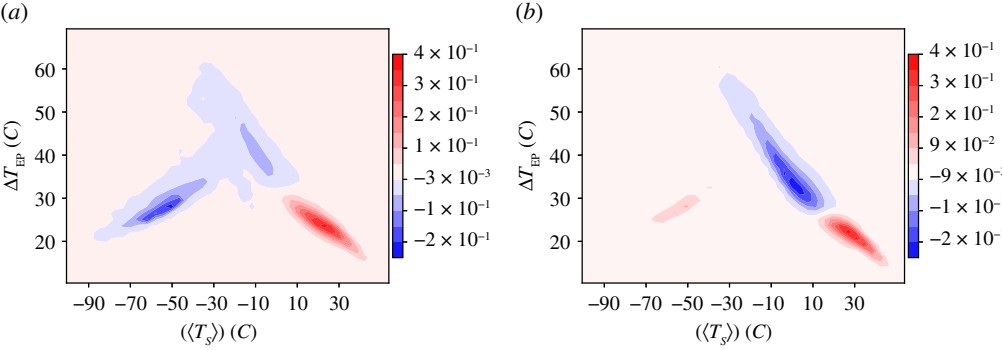

**Figure 4.** First two subdominant eigenvectors of the finite state projected Markov operator for set-up A and $\sigma = 18\%$. (a) First subdominant mode ($\tau_2 \approx 30\,y$) describing the transitions between the two competing metastable states; see also a clear signature of persistent cold departures of the system—within the W basin of attraction—from typical warm conditions leading to the transitions. (b) Second subdominant mode ($\tau_3 \approx 11\,y$) describing the low-frequency variability within the W basin of attraction. Note the lack of time scale separation between these two modes. (Online version in colour.)

### (iii) Relaxation Modes

As detailed in the electronic supplementary material, by constructing a finite-state Markov chain model of the projected ($[\langle T_S \rangle]$, $\Delta T_{EP}$) space, one can extract further useful information about the slow dynamics of the system. We study the statistics of the transitions of the state of the system for the case $\sigma = 18\%$ on a time scale of 30 days. The dominant eigenvector of the Markov chain is the projection of the invariant measure given in figure 3a. The subdominant eigenvectors describes how a generic initial measure relaxes to the invariant one. We remark that, despite the very severe projection, the Markov chain model features positive metric entropy, which measures the rate of creation of information, and positive entropy production, which unequivocally indicates non-equilibrium conditions and is associated with the presence of currents [117].

The two leading subdominant eigenvectors of the finite-state Markov chain approximation of the projection of the 30-days transfer operator in the ($[\langle T_S \rangle]$, $\Delta T_{EP}$) plane for the case $\sigma = 18\%$ are presented in figure 4. The eigenvector shown in panel (a) is associated with the coarse grained, slow process of transition between the two metastable states. The spectral gap of the Markov chain is given by the corresponding eigenvalue $\approx -1/350 = -1/\tau_2$, where $\tau_2 \approx 30\ y$ is the life time of the eigenvector. One of two peaks is negative and the other one is positive, as the mode describes a zero-sum probability transfer. Additionally, this eigenvector has a very clear signature of persistent excursions of the system in the far cold region of the warm attractor. This might be interpreted as a signature of the preferential regions where transitions between the SB and W states take place, compare with figure 3b.

Instead, figure 3b by and large describes the slowest intrawell variability, which takes place in the W basin of attraction: the two closely spaced peaks of opposite sign are on the opposite sides of the peak of the W basin of attraction, with the zero isoline cutting across the peak of the warm attractor; compare with figure 3a. This eigenvector is associated with the process of ice formation and melting and has a lifetime $\tau_3 \approx 11\ y$. A smaller peak is present in correspondence to the SB basin of attraction, indicating that this eigenvector captures some W $\rightarrow$ SB escape process; compare with figure 3b.

## (b) Set-up B: atmospheric-only large-scale energy transport

### (i) The three competing climate states

In [118], it was found that multistability appeared in a non-equilibrium fluid system when considering low values of the viscosity. Here we find something qualitatively similar. Excluding

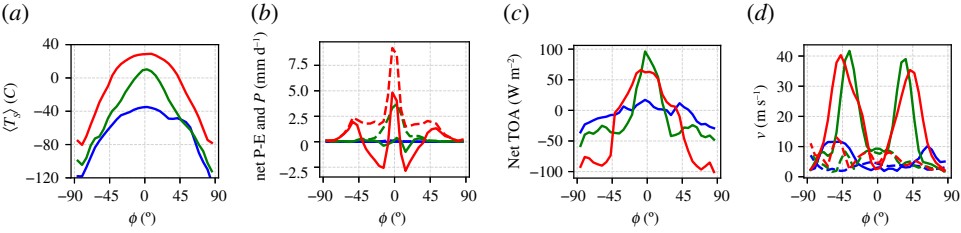

**Figure 5.** Climatology of the zonal averages of (*a*) surface temperature, (*b*) P-E (solid lines) and Precipitation (P, dashed lines), (*c*) TOA net radiation, (*d*) magnitude of zonal wind speed at 300 hPa (solid line) and at 1000 hPa (near surface, dashed lines) versus the latitude $\phi$. Blue lines: SB state. Red lines: W state. Green lines: C state. (Online version in colour.)

the large-scale heat oceanic transport amounts to removing a very powerful negative feedback of mixing, i.e. a mechanism of stabilization for the climate that efficiently redistributes energy throughout the system. This changes qualitatively the global stability properties of the system compared with the case of set-up A. Indeed, in set-up B, using again a large set of initial conditions ranging from very cold to very warm, we find empirical evidence of three competing climate states, whose basic features are reported in table 1, and we refer to the eelctronic supplementary material, figure S2 for further evidence. One of the climates is the fully glaciated SB state, which features very low $\Delta T_{EP}$ and extremely low global temperature, close to $-50°C$. The second climate resembles the W state found in set-up A, featuring an above $0°C$ global temperature, with $\Delta T_{EP} \approx 40°C$ and roughly 27% sea ice coverage. Between the two, lies the C state, which is not fully ice covered, and even though it has $[\langle T_S \rangle] \approx -30 °C$, the fact that $\Delta T_{EP} \approx 50 °C$ suggests the presence of a warm latitudinal band at subtropical latitudes. The presence of an ice-free latitudinal band has huge implications in terms of habitability [30,119]. We remark that such a climate state had not been detected in earlier investigations performed with a virtually identical model set-up [33]. The discovery of the C state has come from considering very unstable initial conditions near the boundary separating the basins of attraction of the W and SB state. Empirically, one discovers that the basin of attraction of the C state is very small compared with those of the SB and W states; see also the electronic supplementary material, figure S3. The quasi-ephemeral nature of the C state becomes clearer when looking at the stochastically perturbed simulations, as discussed later.

In figure 5, we compare the climatology of the three climates (W in red, C in green and SB in blue) resulting from a 40-year average in steady state conditions, in absence of stochastic forcing ($\sigma = 0$). The SB state is very similar to the one obtained with set-up A, as the ocean plays a negligible role in a fully glaciated planet, and will not be further discussed here. The W state is similar with its counterpart in set-up A, albeit considerably colder, and, correspondingly, with a weaker hydrological cycle. We can interpret this as resulting from the ice-albedo feedback. Indeed, the presence of a weaker heat transport towards high latitudes due to removing the action of the ocean leads to a larger sea ice surface—compare figure 2*b* with figure 6*c*—which contributes to lowering the planetary albedo, thus enhancing the input in the energy channel at TOA. Owing to the Boltzmann radiation feedback, the steady state must then be characterized by a lower average temperature compared to set-up A. Finally, the presence of larger temperature differences between high and low latitudes lead to a stronger atmospheric variability, as baroclinic conversion is more efficient and can draw from a larger reservoir of available potential energy. This is associated with a stronger Lorenz energy cycle compared to set-up A, see table 1; see a discussion of the climatic effects of modulating the meridional oceanic heat transport in the W state in [109].

Figure 5*a* shows the climatology of the zonal mean surface temperature. We remark that in the C state the subtropical band $[-20° N, 20° N]$ features above-freezing temperature, while lower temperatures and prevailing sea ice is present at higher latitudes, as shown in figure 2. Despite

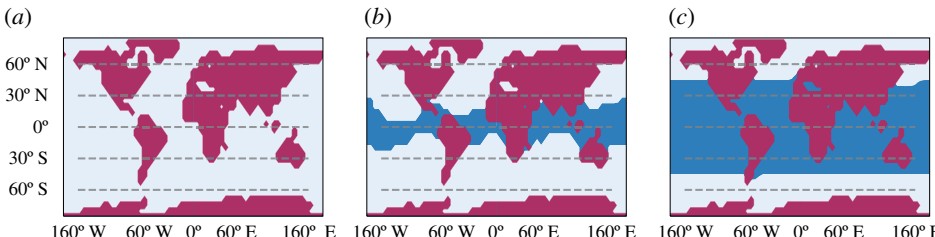

**Figure 6.** Sea ice coverage comparison between (*a*) snowball, (*b*) cold and (*c*) warm climates. Note that the W state of set-up B has more sea ice than the W state of set-up A. (Online version in colour.)

PLASIM's simplified dynamics, the C state shares features of the previously mentioned Slushball state [120] and, especially, of the Jormungand state [34], where the presence of ice-free equatorial band is associated with the dynamics of continental ice sheets and of the interplay of sea ice cover, surface albedo, and atmospheric circulation, respectively. Figure 5*b* shows the zonally averaged P-E and precipitation climatology. The C state features an intense precipitation in the equatorial belt, driven by the strong convection occurring there, but the P-E field indicates that the water vapour is locally recycled and no large-scale transport takes place, as opposed to the W state.

Figure 5*c* shows the zonally averaged net TOA energy budget. One can infer that the meridional atmospheric enthalpy transport has comparable intensity in the W and C climates, yet the peaks of the transport—indicated by vanishing values of the TOA budget [1,107,108]—are confined to lower latitudes in the latter case. This indicates a vigorous heating realized at $\approx \pm 30^\circ N$. Correspondingly, the jet stream for the C state is located at lower latitudes compared to the W climate (figure 5*d*), while it is more intense, as the local meridional temperature gradient throughout the atmosphere is larger. This corresponds to a large temperature difference between low and high latitudes at surface, see table 1.

Finally, the C state features a strong Lorenz energy cycle ($\approx 4.0\,\mathrm{Wm}^{-2}$), thanks to the presence of a large reservoir of available potential energy that can be converted to kinetic energy by baroclinic instability. The intensity of the Lorenz energy cycle of the C state is especially remarkable given that the atmospheric circulation is very weak poleward of 50° latitude.

### (ii) Noise-induced transitions

The presence of three instead of two deterministic attractors makes set-up B considerably more complex than set-up A; for example, now the existence of extra M states connecting SB with C and W with C has to be taken into account, on top of those connecting SB with W already seen in set-up A. Figure 7*a* shows the projection of the invariant measure in the reduced phase space given by ($[\langle T_S \rangle], \Delta T_{\mathrm{EP}}$) obtained for $\sigma = 12\%$, while in figure 7*b* we show the corresponding estimate of the quasi-potential. We remark that in set-up B a lower noise intensity is needed to excite transitions with frequency comparable to what obtained in set-up A, for the basic reason that we are missing the global stabilizing feedback given by the ocean heat transport. This corresponds to having weaker diffusion in the Fokker–Planck operator describing the evolution of probabilities. The location of the deterministic attractors is shown with ellipses of different colour, where magenta, green and cyan correspond to W, C and SB climate states, respectively.

The location of the C state is not directly visible in the projected invariant measure or in the quasi-potential, in the form of a local maximum and minimum, respectively. The operation of performing a projection to such a low-dimensional space is mainly responsible for such a loss of information. This issue is addressed specifically in §4c. Additionally, as we shall see below, the third attractor corresponds to a much shallower local minimum of the quasi-potential compared to the W or SB states. As a result, the C local minimum is (at least in the considered projection) washed out when considering a noise intensity of $\sigma = 12\%$, and it is hard to keep track of orbits

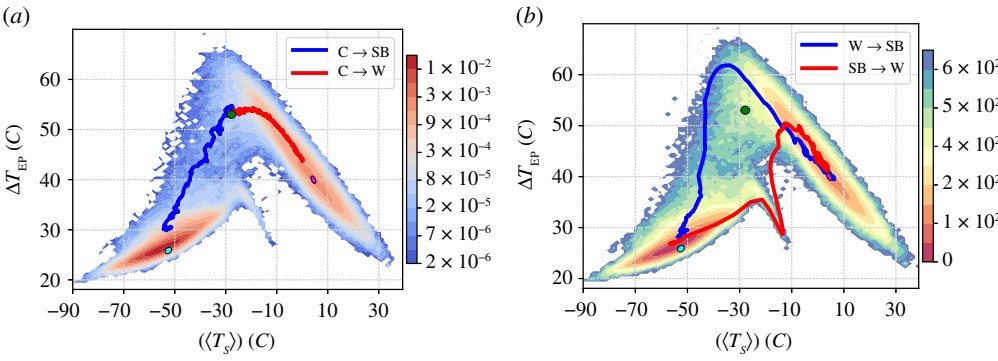

**Figure 7.** Set-up B. (*a*) Two-dimensional projection of the invariant measure on ($[\langle T_S\rangle]$, $\Delta T_{EP}$); $\sigma = 12\%$. The stochastically averaged escapes from the C state with $\sigma = 6\%$ are included. (*b*) Quasi-potential $\Phi$, whose global minimum is set to 0. The blue (red) line indicates the stochastically averaged transition paths for the W→SB (SB→W) transitions. The ellipses indicate the location of the deterministic attractors of the SB (cyan), C (green) and W (magenta) state. (Online version in colour.)

persisting significantly near C, see equations (2.3)–(2.5). This implies the presence of an additional scale relevant for understanding the multistability of the system, along the lines of what discussed in §5. Here one faces a typical dilemma in terms of optimal use of computational resources. Considering a weaker noise would in principle facilitate the detection of the C state, and, in general, of the finer features of the phase space of the system. On the other hand, the exploration of the phase space of the system becomes more difficult, as the stochastic orbit is trapped for a very long time near either the W or the SB state, and the visits to the C state (unless the initial condition is set very close to it, as done below) are extremely unlikely. Hence, it is hard to obtain a reasonably good estimate of the invariant measure given finite computational resources. Note also that, as discussed in [56], in the zero noise limit the invariant measure concentrates on the state featuring the lowest value of the quasi-potential (the SB state, in this case). As discussed in §4c, harnessing methods of data science, and specifically manifold learning, allow us to sort out such a conundrum and to automatically detect with high statistical significance also the C state in the case $\sigma = 12\%$.

As mentioned above, the presence of ocean diffusion triggers the ice-albedo feedback in a direction that favours warming. Accordingly, in set-up B, the minimum of the quasi-potential corresponding to the SB state is deeper than the one corresponding to the W state. This can be seen in figure 8*a*, where the $W \to SB$ and $SB \to W$ mean escapes times are presented as a function of the inverse squared noise amplitude. Using equation (2.5), we obtain the following estimates for the depth of the local quasi-potentials: $\Delta\Phi_{W\to SB} \approx 290(10)$ and $\Delta\Phi_{SB\to W} \approx 500(10)$. As opposed to set-up A, in set-up B the pre-exponential factors of the expectation value of escape times is vastly different. Note that, neglecting the C state, the population of the SB and W state is inversely proportional to the corresponding escape times. As a result, despite being associated with a shallower local minimum of the quasi-potential, the fraction of population in the W state is larger when considering relatively strong noise intensity, whereas eventually, the SB state dominates in the weak-noise limit. Despite the profound dynamical differences between set-up A and B, the estimates of the instantonic and relaxation paths between the SB state and the W state are qualitatively similar; compare figures 3*b* and 7*b*. Furthermore, the interpretation of the different physical mechanisms controlling the SB→W and W→SB transitions paths for set-up B is fundamentally the same as for set-up A.

The more complex geometry of the phase space of set-up B is made apparent by the fact that the transitions between the W and SB states can be either direct or, instead, the paths deviate considerably as the orbit is temporarily trapped near the C state. The reader is encouraged to watch the movies that are linked from the caption of figures S5 and S6 in the electronic

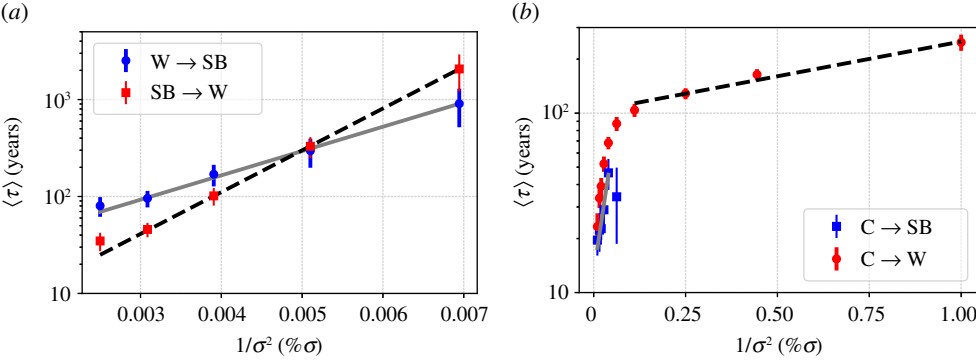

**Figure 8.** Average escape time versus the inverse squared %$\sigma$ in set-up B. (*a*) Comparison of W→SB (filled blue circles) and SB→W (filled red squares) and corresponding exponential fit, grey straight and black dashed lines. (*b*) Comparison of C→SB (filled blue squares) and C→W (filled red circles) and corresponding exponential fit, grey straight and black dashed lines. The fits have been performed using equation (2.5). (Online version in colour.)

supplementary material. Such a trapping is always extremely short-lived compared to the other relevant time scales associated with the transition between the two other metastable states.

The next step is to provide a characterization of the quasi-potential near the C state, and, specifically, to estimate the C→SB and C→W barriers for the local quasi-potential. We then investigate the properties of the system near the C state. Following [61], we bypass the problem of estimating reliably the invariant measure near the C state and investigate the escape process from the C state by considering a large number of independent trajectories initialized in the deterministic C attractor and apply a weaker random forcing with $\sigma = (1\% - 10\%)$. We then collect the statistics of escape times and keep a separate track for trajectories ending up in the W versus in the SB state through the corresponding M states. Using equation (2.5), we are able to estimate the two quasi-potential barriers $\Delta\Phi_{C\to SB}$ and $\Delta\Phi_{C\to W}$. We see in figure 8*b* that $\Delta\Phi_{C\to SB} \approx 16(2)$ (blue filled squares) is about one order of magnitude smaller than the $W \to SB$ and $SB \to W$ barriers. Interestingly, the energy barrier $\Delta\Phi_{C\to W} \approx 0.45(4)$ (red filled circles) turns out to be much smaller than $\Delta\Phi_{C\to SB}$, which explains why below a certain noise level, i.e. $\sigma \approx 4\%$ we practically get no transitions towards the SB attractor, with all escape trajectories ending in the W basin of attraction. Also, for the C→W transitions, we clearly observe from figure 8*b* that for $\sigma$ larger than $\sigma \approx 5\%$ there is a different scaling that can be attributed to the prefactor in equation (2.5), which indicates that the weak-noise limit is not achieved for these values of $\sigma$ for these escape processes. Further comments on the escape trajectories from the C state can be found in the electronic supplementary material.

### (iii) Relaxation modes

Finally, we study the two subdominant eigenvectors of the finite-state Markov chain approximation of the projection of the transfer operator in the $([\langle T_S \rangle], \Delta T_{EP})$ plane for the case $\sigma = 12\%$, see figure 9. As in set-up A, the Markov chain model features positive metric entropy and positive entropy production. We get a broad agreement with the results of set-up A also in terms of interpretation of the meaning of the eigenvectors, but a more clear separation of scales between the two corresponding eigenvalues is present in this case. Figure 9*a* portrays the first subdominant eigenvector. The spectral gap of the Markov chain is given by the corresponding eigenvalue $\approx -1/3500 = -1/\tau_2$, where $\tau_2 \approx 290\,y$ is the life-time of the eigenvector, which matches the life time of the SB state. Because of such a long time scale, and of the fact that the transition time is very short compared to the residence time, we lose any feature of the transition path, as opposed to set-up A. The eigenvector shown in figure 9*b* has a life-time $\tau_3 \approx 10\,y$ and portrays the low-frequency variability in the W basin of attraction, which can lead to occasional transitions

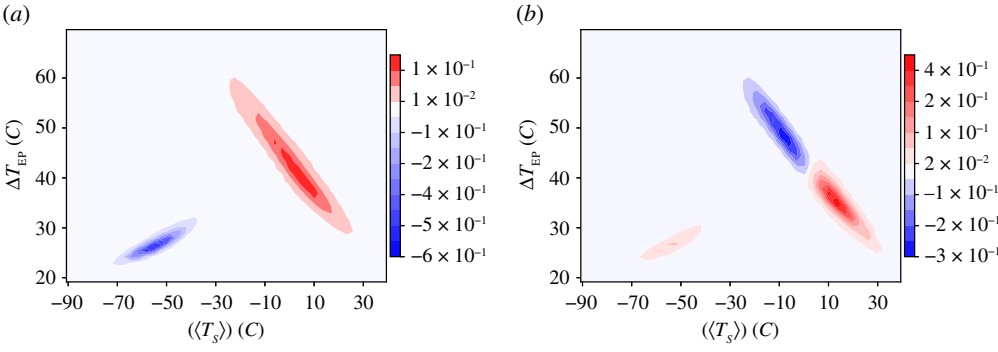

**Figure 9.** First two subdominant eigenvectors of the finite state projected Markov operator for set-up B and $\sigma = 12\%$. (a) First subdominant mode ($\tau_2 \approx 290\,y$) describing the transitions between the W and SB states. (b) Second subdominant mode ($\tau_3 \approx 10\,y$) describing the low-frequency variability within the W state. (Online version in colour.)

towards the SB state; compare the W $\rightarrow$ SB transition path in figure 7b. We find no signature of the presence of the C state, whose life time is much smaller than 10 years for this level of noise. This clarifies that for this level of noise the C state is almost entirely washed out.

## (c) Automatic determination of the metastable states

The basic issue we want to address now is that, while in figure 7 the SB and W state clearly appear as corresponding to local maxima of the projected invariant measure, this is not the case for the C state, in this as well as in many other two-dimensional projections we have tested. Indeed, it has been impossible with the tools developed so far to find any direct evidence of the C state in the stochastic simulation performed with a noise level that was sufficiently strong to allow for the exploration of the full phase space of the system. As described in §4b, the discovery of the C state has been serendipitous and based on the exploration of the phase space via forward deterministic simulations. We next show what can be obtained by applying the suite of data driven methods [72,73,75] presented in §2b to the output of some given numerical simulations taken as pseudo-observations of an in principle unknown model.

We first consider a numerical integration of the model in set-up B lasting $6 \times 10^4$ years and performed with $\sigma = 12\%$. From the complete trajectory of $O(10^5)$ d.f. recorded with having temporal resolution of one time step, we construct a severely coarse-grained version of the phase space by a set of 30-day averaged air temperatures measured every 10 months (hence, decimated with respect to the standard 30-day averaged dataset in previous sections) at three different pressures (300, 500 and 1000 hPa) and 32 different latitudes, for a total of $n = 96$ variables. The quasi-potential as a function of these variables is, in principle, a 96-dimensional function, which cannot be visualized or estimated in a simple manner.

By using the approach outlined in §2b, we study the topography of this function. We first estimate the intrinsic dimension of the manifold containing the data, which turns out to be approximately 11, significantly smaller than the number of variables.[1] This number is approximately scale invariant: indeed the estimated value does not change significantly if the dataset is significantly undersampled. Since the intrinsic dimension of the embedding manifold is relatively low and well defined, one can estimate the quasi-potential $\Phi_t$ in each time frame $t$ using equation (2.7), without defining explicitly the approximately 11 coordinates mapping the manifold. Using these estimates, one finds the candidates for the various attractors, which

---

[1]Note that we should not in any way interpret this number as representative of the actual effective dimension of the attractor of the climate system, because the coarse graining procedure applied in space and time filters out almost entirely the dynamics—which is prevalent in this climate model as well as in reality—occurring over time scales shorter than one season and featuring longitudinally symmetric structure [4].

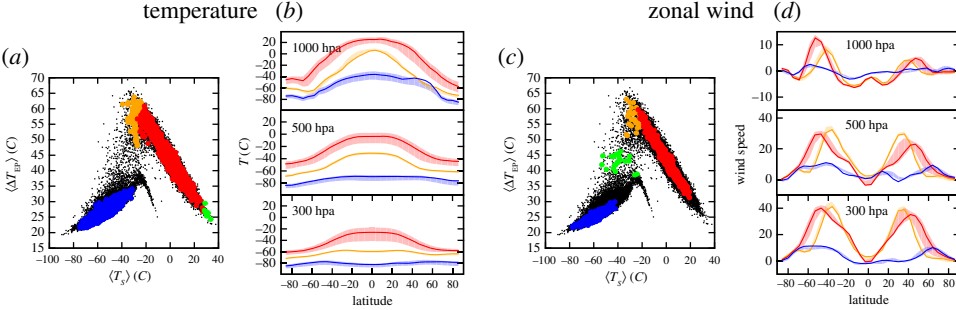

**Figure 10.** The topography of the quasi-potential in high-dimension. Panels (*a,b*): the analysis is performed for set-up B and $\sigma = 12\%$ in the coordinate space of the zonally averaged air temperature at 300, 500 and 1000 hPa at 32 latitudes between $-86°$ and $86°$ (96 variables). Panels (*c,d*): the analysis is performed on the time series of the zonally averaged zonal and the meridional wind, same locations as in (*a,b*). (192 variables). Panels (*a,c*) portrays the ([⟨$T_S$⟩], $\triangle T_{EP}$) projection of the estimated basins of attraction of the quasi-potential. The core sets are coloured in blue (SB state), orange (C state) and red (W state). The green points are the core set of spurious attractors found by the algorithm. The black points are configurations which do not belong to any core state. Panels (*b,d*) portray the average value plus/minus one standard deviation of the variables, restricted to the core sets of the SB (blue), W (red) and C (orange) states, as shaded area. The time averages of the same variables, computed for the corresponding deterministic attractors, is shown in dark solid lines of the same colour, respectively. The meridional wind is not shown. (Online version in colour.)

correspond to the local minima of $\Phi$. With a statistical confidence level of 99%, corresponding to $Z = 2.576$, we find four states, with a core population of 39 171, 12 099, 112 and 11 frames, respectively. The configurations corresponding to the four minima of $\Phi$ were then evolved without stochastic forcing in order to obtain the corresponding asymptotic states, While the first three states are in the basin of attraction of the SB, W and C attractors, correspondingly, the fourth state is found to be unstable, as it forward evolution converges to the W attractor. This indicates that the fourth state is an artifact of finite sampling, or of the variations of the $Z(x)$ (see equation (2.3)), which, in the estimate of $\Phi_t$, are neglected. The configurations assigned to the core set of the three remaining states are represented in figure 10*a* in the same projection used in figure 7. In this projection the C and W states strongly overlap, and no barrier is visible between the two.

In figure 10*b*, we plot the average and the standard deviation, estimated for the core set of each state, of the 96 air temperature variables used in the analysis. Note that such average values agree remarkably well with the time-averages one obtains by considering the corresponding deterministic attractors, represented as continuous lines in figure 10. Remarkably, the distributions are significantly well separated for almost all the variables. This demonstrates that the W and C state are indeed non-overlapping in the 96-dimensional space of these variables. This also shows that the data-driven approach presented here is able to reconstruct accurately the statistical properties of the competing deterministic metastable states.

We have then repeated the exercise by considering the $n = 192$ variables describing the 30-day averaged meridional and zonal wind at the same latitudes and pressure levels as before. The intrinsic dimension of this dataset is approximately 16, slightly larger than for the other variables. In this space, at a statistical confidence of 99% the algorithm can detect only two states, the W and the SB states. At a 98% confidence the C states appears (orange points in figure 10*c*), together with another state, represented in green. The latter state is spurious, since simulations performed with $\sigma = 0$ starting from the estimated minimum rapidly converge to the SB state. In this space the C state is much more similar to the W state, as shown in figure 10*d*: the average zonal wind differs significantly only in the mid-latitudes of the Southern Hemisphere at all levels and in the mid-latitudes of the Northern Hemisphere only at 500 hPa. Note also in this case the excellent agreement obtained with the average statistics computed for the corresponding deterministic attractors.

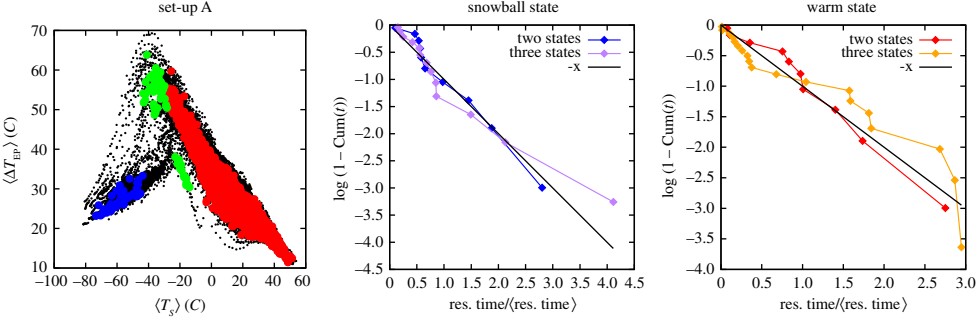

**Figure 11.** The quasi-potential and the residence times for set-up A and $\sigma = 18\%$. Panel ($a$): the states obtained analysing the coordinate space of the air temperatures at three different pressures at 32 latitudes (the same variables used in the analysis in figure 10$a$). The analysis is performed with $Z = 5$. At higher statistical significance the green state disappears. Panels ($b,c$): the empirical cumulative distribution $C(x)$ of the normalized escape time $x = t_{esc}/\langle t_{esc}\rangle$, where $\langle t_{esc}\rangle$ is the average of all the observations. Panel ($b$): the SB state. Panel ($c$): the W state. Blue and red lines: the green state in panel a is not considered meaningful. Purple and orange lines: the green state is considered meaningful. The dashed black lines correspond to the ideal case in which $x \sim \exp(-x)$, and therefore $\log(1 - C(x)) = -x$. (Online version in colour.)

We have also performed the same analysis on a simulation evolved for 32 780 years using the model in set-up A and with $\sigma = 18\%$. At high statistical significance, we detect two states corresponding to the W and the SB climates. At lower statistical significance other states appear, see figure 11$a$. The (spurious) green state occupies a similar regions as the C state found in set-up B, see figure 10$a$. However, the air temperature field is different in the two cases, as the spurious state is consistently colder at all atmospheric levels, even if a good degree of agreement exists in the meridional structure. Correspondingly, a good correspondence between the two states is found when looking at the zonal winds, see figure S9 of the electronic supplementary material. We may then interpret the spurious state as a dynamical remnant—possibly a ghost state [121]—of the C state found for set-up B. Indeed, the spurious state is not an attractor, as it evolves towards the W state if one removes the stochastic forcing. The dynamics of an ensemble of trajectories initiated near the green dots is by and large controlled by two subdominant eigenvectors depicted in figure 4$a,b$.

These results indicate that our approach allows identifying the correct metastable states of a complex high-dimensional dynamic model, but these states come with an uncertainty, which partially derives from statistical errors. If less samples are available, the states will be recognized by a lower statistical confidence, as quantified by the parameter $Z$. For example, if one decimates the frames by a factor four and repeats the analysis in figure 10$a$, the state corresponding to the state corresponding to the C attractor can be detected only at a lower statistical confidence ($Z = 2.3$). Uncertainty also arises from the approximations intrinsic in the quasi-potential estimator, which neglects the pre-exponential factor $Z(x)$. Finally, an error is introduced by the correlation between the frames, which are generated by a dynamic model and sampled with a time lag of a few months. However, one can rather straightforwardly recognize the spurious states, even without performing a simulation at $\sigma = 0$, by estimating, on the same trajectories which brings to their identification, the probability distribution of the first escape times. This distribution is estimated by assuming that the system performs a transition between two states when it visits a core configuration belonging to a state which is different from the state of the last core configuration visited in the past [122]. In this manner, one splits the trajectory in segments, each labelled with a different state, whose length is an estimate of the escape time $t_{esc}$. If the set of states defines (at least approximately) a Markov model, $t_{esc}$ should be exponentially distributed. In figure 11$b,c$, we plot a function of the empirical cumulative probability distribution of $t_{esc}$ which, if $t_{esc} \sim$ Exp, should coincide with the black dashed lines. If one considers as meaningful

also the green state in figure 11*a* one obtains a set of $t_{esc}$ from the W and the SB state whose distribution significantly deviates from an exponential (purple and orange lines figure 11*b,c*). If instead one does not consider the green state as meaningful, the distribution of the escape times from the W and SB state is almost perfectly exponential (blue and red lines), as far as one can judge from the relatively small number of transition events observed in the trajectory. This analysis indicates that our approach allows identifying the *correct* metastable states of the system even from relatively short trajectories, in which only $\mathcal{O}(10)$ transitions are observed. The states can be identified in a fully unsupervised manner, analysing only the trajectory or by running short relaxation dynamics with $\sigma = 0$.

## 5. Conclusion

Achieving a deeper understanding of the nature of the Earth's multistability and related tipping points is one of the key contemporary scientific challenges because it is essential for better framing the co-evolution of climatic conditions and of the biosphere throughout the Earth's history, and, in the present context, for better constraining the current planetary boundaries through a careful examination of the safe operating space for humanity [123].

Systems undergoing stochastic dynamics and featuring competing multistable states can be effectively described by taking advantage of the formalism of the quasi-potential landscape, which generalizes the notion of the free energy to non-equilibrium systems. Local minima in the quasi-potential describe competing metastable states, and are separated by local maxima and saddles—M states—that define possible gateways for transitions. To demonstrate our framework in the case of the climate we employ two versions of an open source climate model, PLASIM, which has an appropriate mix of precision, flexibility and efficiency in simulating the present climate as well as very exotic climatic conditions. The first version (set-up A) features a simplified but meaningful representation of the oceanic energy transport from low to high latitudes, whereas in the second one (set-up B) large-scale energy transport is provided solely by the turbulent atmosphere. Set-up A demonstrates the well-known competing climatic states corresponding to the present warm (W) conditions and the so-called snowball (SB) climate. Set-up B, instead, contains an unexpected additional intermediate stable climate (C) where the sea is partially ice-free in the equatorial band. The lack of a powerful mechanism of energy redistribution across the climate makes this additional state possible. Despite PLASIM's relative simplicity, the C state should not be regarded as a pure mathematical curiosity corresponding to a pathological solution: exotic climate states rather similar to the C state obtained here have been obtained in other climate models and are deemed extremely relevant in paleoclimatic terms because they provide a scenario able to explain the survival of life during the Neoproterozoic glaciations.

The phase space of the model can be explored when stochastic forcing—here in the form of a yearly fluctuating solar irradiance—is introduced, leading to transitions between the competing metastable states. We compute the quasi-potential function, which describes, on the one side, the invariant measure of the system and, on the other side, in its local version, controls the probability of transition of the stochastically forced trajectory from one to another basin of attraction. We are able to estimate in both set-ups the optimal escape paths—the instantons—and the corresponding relaxation trajectories linking the W and SB states, and are then able to verify the non-equivalence between the two, which is an essential feature of non-equilibrium properties.

Instantons describe how transitions take place in the zero-noise limit and are more of a mathematically elegant construction than a physically relevant object in our investigations, as we need to consider noise of moderate yet non-negligible intensity in order to observe reasonably frequent transitions between the SB and W attractors. Additionally, studying the transfer operator in a suitably projected space sheds light on how the system relaxes to its invariant measure. We are able to find clear evidence of both interwell relaxation processes, which describe transitions between competing metastable states, and are the noisy version of instantons, and intrawell relaxation processes, which would conventionally be labelled as ultralow frequency variability

within the W state associated with large-scale melting and thawing of sea ice and corresponding large temperature fluctuations.

A non-trivial result we obtain is that the instantons escaping the SB and the W attractors do not meet at one of the M states separating the two corresponding basins of attraction. This can be best appreciated visually by watching the movies included in the electronic supplementary material. In fact, the transitions take place through two separate saddles. This has two important implications (a) the dynamics on the basin boundary is, by itself, multistable; and (b) one has large-scale non-vanishing currents in the phase space. This is a strong signature of the non-equilibrium nature of the system. The existence of separate paths for the SB-to-W and W-to-SB states marks a relevant difference with previous studies. The presence of more evident macroscopic signature of non-equilibrium conditions can be attributed to the presence in this model of an active hydrological cycle, which is the major agent of entropy production in the climate system.

The C state in set-up B corresponds to a comparably shallower minimum of the quasi-potential, which can be explored only considering significantly weaker noise than needed to explore globally the phase space of the system. We discover that the most natural, preferential escape route from the C state is towards the W state. The C state is only barely metastable, as even internally generated noise of the numerical discretization can destabilize it, even if only rarely and over ultra long time scales, as discussed in the ESM. The position in phase space of the C state and its properties indicate that it is likely that the C state is the leftover of the M state between the SB and W climate obtained as we progressively switch off the horizontal diffusivity of the ocean, because this leads to a less efficient redistribution of energy in the system,

We have complemented the top-down approach based on numerical modelling with bottom-up data-driven methods that allow for the automatic detection of the competing metastable states from the analysis of a single long stochastic trajectory and to reconstruct the quasi-potential using an arbitrarily high-dimensional input dataset. Using this approach, we have been able to reconstruct the dynamical landscape of the climate model in both set-ups and gain a better understanding of how transitions between the competing metastable states occur. Remarkably, by suitable averaging over many realizations, we have been able to reconstruct the climates of the competing (deterministic) metastable states.

## (a) Outlook: multiscale multistability

The quasi-potential landscape viewpoint might provide a useful way for describing the multistability of the climate in a hierarchical fashion. We present in figure 12$a$ an illustration of this perspective, where the possible states of the climate are described by the vector $X$. The quasi-potential $\Phi$ features troughs, saddles and ridges at different scales.

The intensity of noise allowing for exploring transitions between competing states decreases dramatically as we go from level 1 to level 3, because the local minima become shallower. Going to even smaller scales, one would find additional (shallower) corrugations of $\Phi$. Multistability in the climate system is often revealed by the presence of hysteresis loops obtained when suitable parameters of the system are changed, usually quasi-adiabatically [33,39,124]. Figure 12$b$ shows schematically how the multistability portrayed in figure 12$a$ appears when applying suitable protocols of parametric modulations to the system.

The above description could potentially be a fundamental mathematical structure linking the global multistability of the climate system with the geographically localized tipping elements and the so-called cascading tipping points, and might be useful for understanding the associated multiscale hysteretic behaviour of the climate system when parameters are suitably modulated. We stress that in the current work we have been able to explore only the highest hierarchical level of multistability. A more complete climate model and a suitable, different choice of stochastic forcing would be needed for exploring the small-scale local minima of the quasi-potential associated, e.g. with competing climate states that exchange stability at tipping points like the ocean associated with the AMOC shutdown. In this case, one would need a model able to resolve

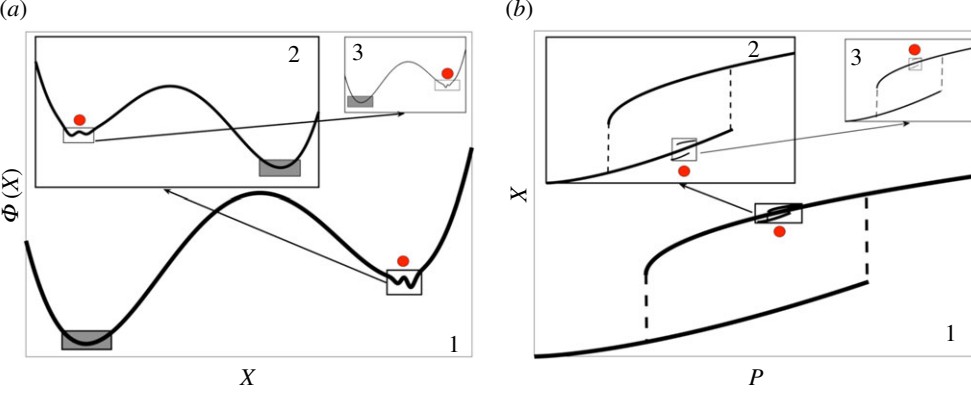

**Figure 12.** Schematic representation of the multiscale nature of multistability in the climate system. (*a*) Quasi-potential $\Phi$ as a function of the state of the system $X$. (*b*) Corresponding hysteresis loops as a function of a parameter $P$. The white boxes indicate the zoomed-in current state of the system (red dot), going from 1 to 3 towards smaller and smaller scales. (Online version in colour.)

explicitly the large-scale ocean circulation and possibly consider random perturbations to the hydrological cycle acting in the North Atlantic sector.

We envision the combination of the top-down and bottom-up approach as a possible way forward to study the multiscale nature of the multistability of the climate system, as well as of other systems of comparable complexity. This research work paves the way for further investigation into some fundamental properties of the climate system and goes in the direction of clarifying its intransitive versus quasi-transitive versus transitive nature [125] when different time scales are considered. Additionally, it indicates a way for fostering the development of climate models of different level of complexity: indeed, we want them to be able to capture the qualitative features of climate, by allowing for the presence of a complex dynamical landscape featuring hierarchically arranged—according to the desired level of envisaged detail and granularity—competing metastable states, associated with the ensuing tipping points.

The viewpoint presented here seems also promising for investigating a separate, extremely relevant aspect of atmospheric dynamics, namely the existence in the atmosphere of different regimes of operation, which define the presence of substantial low-frequency variability on subseasonal time scales [4,126]. This boils down to the fact that, at coarse-grained level, due to extreme dynamical heterogeneity [127], one is practically looking at a multistable system, where one can define and detect transitions between different metastable states [128].

Finally, we remark that white Gaussian noise might not necessarily be the only suitable way to treat stochasticity in the climate system [129]. The theory of escapes from attractors in the presence of Lévy noise has been developed [130,131] and very recently applied to simple geophysical models [132]. It is well known that the mechanisms of escape are rather different than in the standard Gaussian scenario pursued in this paper. It seems then of great relevance to consider the effect of Lévy noise forcing in a more complex climate model like the one considered here.

Data accessibility. The data required to generate the figures can be accessed in the compressed folder that can be found at https://doi.org/10.6084/m9.figshare.c.5431597. The movies presenting examples of transitions are publicly available on the youtube.com platform through the links that can be found in the electronic supplementary material, which is also available at https://doi.org/10.6084/m9.figshare.13079489.

Authors' contributions. G.M. performed the simulations, contributed to the data analysis and to the writing of the paper. T.G. contributed to the writing of the paper. A.L. contributed to the data analysis and to the writing of the paper. V.L. proposed the research topic, contributed to the interpretation of the data analysis and led the writing of the paper. All authors gave final approval for publication and agree to be held accountable for the work performed therein.

Competing interests. We declare we have no competing interests.

**Funding.** T.G. acknowledges the support received from the EPSRC project EP/T011866/1. V.L. acknowledges the support received from the EPSRC project EP/T018178/1. V.L. and G.M. acknowledge the support received from the EU Horizon 2020 project TiPES (grant no. 820970).

**Acknowledgements.** V.L. wishes to thank T. Bódai, N. Boers, M. Ghil, F. Lunkeit, G. Pavliotis, A. Tantet, T. Tel, J. Yorke, and N. Zagli for many inspiring conversations on multistability and tipping points. G.M. wishes to thank F. Lunkeit for his guidance on PLASIM and kind hospitality at the University of Hamburg.

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
