## [Peer Review File · Proceedings. Mathematical, Physical, and Engineering Sciences]

Review History

RSPA-2021-0019.R0 (Original submission)

Review form: Referee 1

Is the manuscript an original and important contribution to its field?

Acceptable

Is the paper of sufficient general interest?

Good

Is the overall quality of the paper suitable?

Good

Can the paper be shortened without overall detriment to the main message?

Yes

Do you think some of the material would be more appropriate as an electronic appendix?

No

Do you have any ethical concerns with this paper?

No

Recommendation?

Accept with minor revision (please list in comments)

Comments to the Author(s)

Report for "Dynamical Landscape and Multistability of a Climate Model" by Margazoglou et al

This paper numerically analyses the multistability of the intermediate complexity model PLASIM. The authors study the multi stability via first long time trajectories, and then show that they can determine the meta stable states via a quasi-potential and a data driven manifold learning procedure. By means of averaging they find the most likely transition states. PLAISM is run in two different set-ups which exhibits different global dynamics. The authors discuss the outcomes of their analysis and provide nice physical interpretations of their results.

I enjoyed reading the paper and have only some minor comments the authors may want to take on board:

** A brief discussion providing some heuristics for Eqn (2.8) would be helpful to the reader and would help appreciating the authors approach.

** How sensitive are the results on the amount of data available and the sampling rates? Some discussion would be good.

** For both set-ups, A and B, the authors perform deterministic long time simulations for each of the meta stable states. There are no details given, however, on the choice of the initial conditions. How were they chosen? Randomly? Educated guesses?

** The authors use the abbreviation TOA. I assume it means "top of the atmosphere" but it should be properly defined.

** the authors find that the instantons connecting the cold and the warm state (and vice versa) do not meet at the M state. They suggest that this is due to the projection they chose, which sounds plausible. Does this dependent on the noise strength σ and for sufficiently large σ , one would see that the two instantons meet up?

** for the results on the almost invariant sets of the Markov chain, I would like to have more details. Can the authors report on the 2nd and 3rd (and 4th) eigenvalue, as well as on the spectral gap (if there is any).

** for the Set-up B the authors find a third metastable state. In (b) they state that the noise level they chose might be too large and has "washed out" the C state. What about smaller noise levels then? Why not show results for that?

** In Appendix 4 on the Transfer Operator, the authors may want to cite standard literature on Ulam's method rather than citing their own papers which would give a wrong impression.

Review form: Referee 2

Is the manuscript an original and important contribution to its field?

Good

Is the paper of sufficient general interest?

Good

Is the overall quality of the paper suitable?

Excellent

Can the paper be shortened without overall detriment to the main message?

Yes

Do you think some of the material would be more appropriate as an electronic appendix?

No

Do you have any ethical concerns with this paper?

No

Recommendation?

Accept with minor revision (please list in comments)

Comments to the Author(s)

The paper "Dynamical Landscape and Multistability of a Climate Model" by Georgios Margazoglou, Tobias Grafke, Alessandro Laio and Valerio Lucarini is a very interesting manuscript on the multistability in an intermediate complex climate model (PLASIM). By varying the solar constant, the authors find either two stable states in the default setup of PLASIM (including oceanic heat transport) or three stable states in a second setup of PLASIM (excluding oceanic heat transport). Although PLASIM does not explicitly model the deep ocean circulation, I consider the results of significant importance. The paper is well written and the results can be followed well due to the elaborated explanations. The results are original in my opinion. Also, the supplied video material and the supplementary information is well documented and can be followed easily. I also appreciated reading the nice conclusion and outlook, which are elaborating on limitations and future directions of research (such as the effects of non-Gaussian noise).

Therefore, I would like to recommend the paper for publication in Proceedings of the Royal Society A after the following minor revisions have been performed.

Major comments:

- 1) Page 16/17 - Figure 9: I had difficulties to understand why the authors chose a sigma value (12%) that makes the detection of the cold-state (C-state) impossible to view (see Fig. 9), while lower sigma-values have been used to construct Fig. 8. I think one or two panels could be added to Figure 9, which also show the C-state, maybe using a lower sigma-value or a shorter lifetime. If this is not feasible, I would ask the authors to more clearly explain that to the reader in section (iii) on page 16/17.
- 2) Page 19, line 39-41: The authors write "However, the distribution of the air temperature variables in this state differs significantly from the C state in setup B (not shown)". Although, it is clearly disentangled what the detected green state is and what its difference to the C state is, I would feel it would be better to show the air temperature of the green state and the C state in a supplementary Figure (instead of writing "not shown").

Minor comments (including some typos):

- 1) Abstract: The authors write "... in one of the two climate models ...", but actually only one climate model (PLASIM) is used in two different setups. This should be clarified. Also, the name of the model could be named in the abstract
- 2) Page 2, line 28: The authors could consider citing another EMIC apart from the CLIMBER-2/3alpha family (e.g. Bern3D, Loveclim, ...)
- 3) Page 3, line 11: A full stop is missing after source "..., 49)"
- 4) Page 3, line 37: A comma should replace the point after source "...65)."
- 5) Page 4, line 28: "asymptotic states"  "asymptotic state"
- 6) Page 8, line 28/29: The authors state that PLASIM cannot represent the process of deep water formation. Do the authors have an intuition of how the results might change in case the model could do that?

Decision letter (RSPA-2021-0019.R0)

20-Apr-2021

Dear Dr Lucarini,

On behalf of the Editor, I am pleased to inform you that your Manuscript RSPA-2021-0019 entitled "Dynamical Landscape and Multistability of a Climate Model" has been accepted for publication subject to minor revisions in Proceedings A. Please find the referees' comments below.

The reviewer(s) have recommended publication, but also suggest some minor revisions to your manuscript. Therefore, I invite you to respond to the reviewer(s)' comments and revise your manuscript. Please note that we have a strict upper limit of 28 pages for each paper. Please endeavour to incorporate any revisions while keeping the paper within journal limits. Please note that page charges are made on all papers longer than 20 pages. If you cannot pay these charges you must reduce your paper to 20 pages before submitting your revision. Your paper has been ESTIMATED to be 28 pages. We cannot proceed with typesetting your paper without your agreement to meet page charges in full should the paper exceed 20 pages when typeset. If you have any questions, please do get in touch.

It is a condition of publication that you submit the revised version of your manuscript within 7 days. If you do not think you will be able to meet this date please let me know in advance of the due date.

To revise your manuscript, log into <https://mc.manuscriptcentral.com/prsa> and enter your Author Centre, where you will find your manuscript title listed under "Manuscripts with Decisions." Under "Actions," click on "Create a Revision." Your manuscript number has been appended to denote a revision.

You will be unable to make your revisions on the originally submitted version of the manuscript. Instead, revise your manuscript and upload a new version through your Author Centre.

When submitting your revised manuscript, you will be able to respond to the comments made by the referee(s) and upload a file "Response to Referees" in Step 1: "View and Respond to Decision Letter". You can use this to document any changes you make to the original manuscript. In order to expedite the processing of the revised manuscript, please be as specific as possible in your response to the referee(s).

IMPORTANT: Your original files are available to you when you upload your revised manuscript. Please delete any redundant files before completing the submission process.

When uploading your revised files, please make sure that you include the following as we cannot proceed without these:

- 1) A text file of the manuscript (doc, txt, rtf or tex), including the references, tables (including captions) and figure captions. Please remove any tracked changes from the text before submission. PDF files are not an accepted format for the "Main Document".
- 2) A separate electronic file of each figure (tif, eps or print-quality pdf preferred). The format should be produced directly from original creation package, or original software format.
- 3) Electronic Supplementary Material (ESM): all supplementary materials accompanying an accepted article will be treated as in their final form. Note that the Royal Society will not edit or typeset supplementary material and it will be hosted as provided. Please ensure that the

supplementary material includes the paper details where possible (authors, article title, journal name). Supplementary files will be published alongside the paper on the journal website and posted on the online figshare repository (<https://figshare.com>). The heading and legend provided for each supplementary file during the submission process will be used to create the figshare page, so please ensure these are accurate and informative so that your files can be found in searches. Files on figshare will be made available approximately one week before the accompanying article so that the supplementary material can be attributed a unique DOI. Alternatively you may upload a zip folder containing all source files for your manuscript as described above with a PDF as your "Main Document". This should be the full paper as it appears when compiled from the individual files supplied in the zip folder.

Article Funder

Please ensure you fill in the Article Funder question on page 2 to ensure the correct data is collected for FundRef (<http://www.crossref.org/fundref/>).

Media summary

Please ensure you include a short non-technical summary (up to 100 words) of the key findings/importance of your paper. This will be used for to promote your work and marketing purposes (e.g. press releases). The summary should be prepared using the following guidelines:

*Write simple English: this is intended for the general public. Please explain any essential technical terms in a short and simple manner.

*Describe (a) the study (b) its key findings and (c) its implications.

*State why this work is newsworthy, be concise and do not overstate (true 'breakthroughs' are a rarity).

*Ensure that you include valid contact details for the lead author (institutional address, email address, telephone number).

Cover images

We welcome submissions of images for possible use on the cover of Proceedings A. Images should be square in dimension and please ensure that you obtain all relevant copyright permissions before submitting the image to us. If you would like to submit an image for consideration please send your image to proceedingsa@royalsociety.org

Open Access

You are invited to opt for open access, our author pays publishing model. Payment of open access fees will enable your article to be made freely available via the Royal Society website as soon as it is ready for publication. For more information about open access please visit <https://royalsociety.org/journals/authors/open-access/>. The open access fee for this journal is £1700/\$2380/€2040 per article. VAT will be charged where applicable. Please note that if the corresponding author is at an institution that is part of a Read and Publishing deal you are required to select this option. See <https://royalsociety.org/journals/librarians/purchasing/read-and-publish/read-publish-agreements/> for further details.

Once again, thank you for submitting your manuscript to Proceedings A and I look forward to receiving your revision. If you have any questions at all, please do not hesitate to get in touch.

Best wishes
Raminder Shergill
proceedingsa@royalsociety.org
Proceedings A

on behalf of
 Professor Johannes Zimmer
 Board Member
 Proceedings A

Reviewer(s)' Comments to Author:

Referee: 1

Comments to the Author(s)

Report for "Dynamical Landscape and Multistability of a Climate Model" by Margazoglou et al

This paper numerically analyses the multistability of the intermediate complexity model PLASIM. The authors study the multi stability via first long time trajectories, and then show that they can determine the meta stable states via a quasi-potential and a data driven manifold learning procedure. By means of averaging they find the most likely transition states. PLAISM is run in two different set-ups which exhibits different global dynamics. The authors discuss the outcomes of their analysis and provide nice physical interpretations of their results.

I enjoyed reading the paper and have only some minor comments the authors may want to take on board:

** A brief discussion providing some heuristics for Eqn (2.8) would be helpful to the reader and would help appreciating the authors approach.

** How sensitive are the results on the amount of data available and the sampling rates? Some discussion would be good.

** For both set-ups, A and B, the authors perform deterministic long time simulations for each of the meta stable states. There are no details given, however, on the choice of the initial conditions. How were they chosen? Randomly? Educated guesses?

** The authors use the abbreviation TOA. I assume it means "top of the atmosphere" but it should be properly defined.

** the authors find that the instantons connecting the cold and the warm state (and vice versa) do not meet at the M state. They suggest that this is due to the projection they chose, which sounds plausible. Does this dependent on the noise strength σ and for sufficiently large σ , one would see that the two instantons meet up?

** for the results on the almost invariant sets of the Markov chain, I would like to have more details. Can the authors report on the 2nd and 3rd (and 4th) eigenvalue, as well as on the spectral gap (if there is any).

** for the Set-up B the authors find a third metastable state. In (b) they state that the noise level they chose might be too large and has "washed out" the C state. What about smaller noise levels then? Why not show results for that?

** In Appendix 4 on the Transfer Operator, the authors may want to cite standard literature on Ulam's method rather than citing their own papers which would give a wrong impression.

Referee: 2

Comments to the Author(s)

The paper "Dynamical Landscape and Multistability of a Climate Model" by Georgios Margazoglou, Tobias Grafke, Alessandro Laio and Valerio Lucarini is a very interesting

manuscript on the multistability in an intermediate complex climate model (PLASIM). By varying the solar constant, the authors find either two stable states in the default setup of PLASIM (including oceanic heat transport) or three stable states in a second setup of PLASIM (excluding oceanic heat transport). Although PLASIM does not explicitly model the deep ocean circulation, I consider the results of significant importance. The paper is well written and the results can be followed well due to the elaborated explanations. The results are original in my opinion. Also, the supplied video material and the supplementary information is well documented and can be followed easily. I also appreciated reading the nice conclusion and outlook, which are elaborating on limitations and future directions of research (such as the effects of non-Gaussian noise).

Therefore, I would like to recommend the paper for publication in Proceedings of the Royal Society A after the following minor revisions have been performed.

Major comments:

1) Page 16/17 – Figure 9: I had difficulties to understand why the authors chose a sigma value (12%) that makes the detection of the cold-state (C-state) impossible to view (see Fig. 9), while lower sigma-values have been used to construct Fig. 8. I think one or two panels could be added to Figure 9, which also show the C-state, maybe using a lower sigma-value or a shorter lifetime. If this is not feasible, I would ask the authors to more clearly explain that to the reader in section (iii) on page 16/17.

2) Page 19, line 39-41: The authors write “However, the distribution of the air temperature variables in this state differs significantly from the C state in setup B (not shown)”. Although, it is clearly disentangled what the detected green state is and what its difference to the C state is, I would feel it would be better to show the air temperature of the green state and the C state in a supplementary Figure (instead of writing “not shown”).

Minor comments (including some typos):

- 1) Abstract: The authors write “... in one of the two climate models ...”, but actually only one climate model (PLASIM) is used in two different setups. This should be clarified. Also, the name of the model could be named in the abstract
- 2) Page 2, line 28: The authors could consider citing another EMIC apart from the CLIMBER-2/3alpha family (e.g. Bern3D, Loveclim, ...)
- 3) Page 3, line 11: A full stop is missing after source “..., 49)”
- 4) Page 3, line 37: A comma should replace the point after source “...,65).”
- 5) Page 4, line 28: “asymptotic states”  “asymptotic state”
- 6) Page 8, line 28/29: The authors state that PLASIM cannot represent the process of deep water formation. Do the authors have an intuition of how the results might change in case the model could do that?

Board Member:

Comments to Author(s):

The reviewers raise several points to improve the manuscript; these points seem fair to me. Please address them in a revision as far as possible and sensible.

Author's Response to Decision Letter for (RSPA-2021-0019.R0)

See Appendix A.

Decision letter (RSPA-2021-0019.R1)

04-May-2021

Dear Dr Lucarini

I am pleased to inform you that your manuscript entitled "Dynamical Landscape and Multistability of a Climate Model" has been accepted in its final form for publication in Proceedings A.

Our Production Office will be in contact with you in due course. You can expect to receive a proof of your article soon. Please contact the office to let us know if you are likely to be away from e-mail in the near future. If you do not notify us and comments are not received within 5 days of sending the proof, we may publish the paper as it stands.

As a reminder, you have provided the following 'Data accessibility statement' (if applicable). Please remember to make any data sets live prior to publication, and update any links as needed when you receive a proof to check. It is good practice to also add data sets to your reference list. Statement (if applicable): The data required to generate the figures can be accessed via this repository: <https://doi.org/10.6084/m9.figshare.13079489>. Animations presenting examples of transitions are publicly available on the youtube.com platform through the links that can be found in the text that can be found at <https://doi.org/10.6084/m9.figshare.13079489>.

Open access

You are invited to opt for open access, our author pays publishing model. Payment of open access fees will enable your article to be made freely available via the Royal Society website as soon as it is ready for publication. For more information about open access please visit <https://royalsociety.org/journals/authors/which-journal/open-access/>. The open access fee for this journal is £1700/\$2380/€2040 per article. VAT will be charged where applicable.

Note that if you have opted for open access then payment will be required before the article is published – payment instructions will follow shortly.

If you wish to opt for open access then please inform the editorial office (proceedingsa@royalsociety.org) as soon as possible.

Your article has been estimated as being 28 pages long. Our Production Office will inform you of the exact length at the proof stage.

Proceedings A levies charges for articles which exceed 20 printed pages. (based upon approximately 540 words or 2 figures per page). Articles exceeding this limit will incur page charges of £150 per page or part page, plus VAT (where applicable).

Under the terms of our licence to publish you may post the author generated postprint (ie. your accepted version not the final typeset version) of your manuscript at any time and this can be made freely available. Postprints can be deposited on a personal or institutional website, or a recognised server/repository. Please note however, that the reporting of postprints is subject to a media embargo, and that the status the manuscript should be made clear. Upon publication of the definitive version on the publisher's site, full details and a link should be added.

You can cite the article in advance of publication using its DOI. The DOI will take the form: 10.1098/rspa.XXXX.YYYY, where XXXX and YYYY are the last 8 digits of your manuscript number (eg. if your manuscript number is RSPA-2017-1234 the DOI would be 10.1098/rspa.2017.1234).

For tips on promoting your accepted paper see our blog post:
<https://royalsociety.org/blog/2020/07/promoting-your-latest-paper-and-tracking-your-results/>

On behalf of the Editor of Proceedings A, we look forward to your continued contributions to the Journal.

Sincerely,
Raminder Shergill
proceedingsa@royalsociety.org

Appendix A

Reply to the referees of the paper RSPA-2021-0019 “Dynamical Landscape and Multistability of a Climate Model”

Dear Editor,

Please find below out detailed answers to the comments and criticisms of the reviewers, which we have found of great usefulness for improving the quality and clarity of our paper.

We are thankful for the overall positive evaluation of our work.

We report below in black the itemized comments of the reviewers, followed, in red, by our replies.

All the very best,

Valerio Lucarini
(for all authors)

Referee 1.

Remark 1: “A brief discussion providing some heuristics for Eqn (2.8) would be helpful to the reader and would help appreciating the authors approach.”

Reply to remark 1:

We added a sentence after eq 2.7 (previously equation 2.8) illustrating the special case $a=0$, in which the estimator reduces to an intuitive and well known form. Specifically, In this case, the quasipotential is estimated as minus the logarithm of the density estimated by a standard k-NN estimator. The a -dependent term allows correcting for linear variations of the density in the neighborhood

Remark 2: “How sensitive are the results on the amount of data available and the sampling rates? Some discussion would be good.”

Reply to remark 2:

Regarding the data-driven method of Sec. (c), as it is discussed, we get an improved determination of the metastable states as the noise level decreases, because less spurious states appear. We added the following sentence:

If less samples are available, the states will be recognized by a lower statistical confidence, as quantified by the parameter dZ . For example, if one decimates the frames by a factor four and repeats the analysis in Fig 10(a), the state corresponding to the state corresponding to the C attractor can be detected only at a lower statistical confidence ($Z=2.3$).

Remark 3: “For both set-ups, A and B, the authors perform deterministic long time simulations for each of the metastable states. There are no details given, however, on the choice of the initial conditions. How were they chosen? Randomly? Educated guesses?”

Reply to remark 3:

Thank you for the comment. We have now added the following sentences.

For Setup A at page 8:

Using a large set of initial conditions ranging from very cold to very warm, we have found empirical evidence of (only) two competing asymptotic states corresponding to the W and SB climates, in agreement with a plethora of previous investigations, as discussed in the introduction.

For Setup B at page 13:

Indeed, in setup B, using again a large set of initial conditions ranging from very cold to very warm, we find empirical evidence of three competing climate states ...

Below, referring to the C state at page 13:

We remark that such a climate state had not been detected in earlier investigations performed with a virtually identical model setup (27). The discovery of the C state has come from considering very unstable initial conditions near the boundary separating the basins of attraction of the W and SB state. Empirically, one discovers that the basin of attraction of the C state is very small compared to those of the SB and W states; see also Fig. 3 in the SM. The quasi-ephemeral nature of the C state becomes clearer when looking at the stochastically perturbed simulations, as discussed later.

Remark 4: “The authors use the abbreviation TOA. I assume it means” top of the atmosphere” but it should be properly defined.”

Reply to remark 4:

We thank the reviewer for pointing us to this issue. TOA has been spelled out as top-of-the-atmosphere.

Remark 5: “the authors find that the instantons connecting the cold and the warm state (and vice versa) do not meet at the M state. They suggest that this is due to the projection they

chose, which sounds plausible. Does this depend on the noise strength σ and for sufficiently large σ , one would see that the two instantons meet up?”

Reply to remark 5:

The reviewer is correct in stating that we find the instanton does not meet at the M state, which is indeed an interesting outcome of this work. Indeed we find that none of the instantons connecting the two or three basins (setup A or B, respectively) intersect in a mutual Melancholia state, contrary to previous observations (i.e. Lucarini & Bodai, 2019, PRL). This is remarkable since it hints towards the strong nonequilibrium nature of the problem. Note though that this cannot be because of the projection, since if the trajectories do not cross in any projection, they cannot cross in the full state space. It is instead the other way around: The projection into the 2d plane makes it look as if the instantons are crossing at a point, which the 3d plot in Fig. 3(d) reveals to be an artifact. We changed the wording on page 10 to clarify this point.

Remark 6: “for the results on the almost invariant sets of the Markov chain, I would like to have more details. Can the authors report on the 2nd and 3rd (and 4th) eigenvalue, as well as on the spectral gap (if there is any).”

Reply to remark 6:

We thank the reviewer for asking us to clarify these points. The second and third eigenvectors of the Markov chain in the projected space are already reported in setup A (fig 4) and B (fig. 9). In the text we have now reported the spectral gap and related it to the life-time of the first subdominant eigenvector, see Pages 12 and 17. The fourth eigenvectors do not convey much extra information (at least for the qualitative scope pursued) and are not reported.

Remark 7: “for the Set-up B the authors find a third metastable state. In (b) they state that the noise level they chose might be too large and has “washed out” the C state. What about smaller noise levels then? Why not show results for that?”

Reply to remark 7:

This issue is now addressed at page 15 before figure 7. We have added a few sentences as follows:

Here one faces a typical dilemma in terms of optimal use of computational resources. Considering a weaker noise would in principle facilitate the detection of the C state, and, in general, of the finer features of the phase space of the system. On the other hand, the exploration of the phase space of the system becomes more difficult, as the stochastic orbit is trapped for a very long time near either the W or the SB state, and the visits to the C state (unless the initial condition is set very close to it, as done below) are extremely unlikely. Hence, it is hard to obtain a reasonably good estimate of the invariant measure given finite computational resources. Note also that, as discussed in [50], in the zero noise limit the invariant measure concentrates on the state featuring the lowest value of the quasi-potential

(the SB state, in this case). As discussed below in Sect. 4(c), harnessing methods of data science, and specifically manifold learning, allow us to sort out such a conundrum and to automatically detect with high statistical significance also the C state in the case $\sigma = 12\%$

Remark 8: “In Appendix 4 on the Transfer Operator, the authors may want to cite standard literature on Ulam’s method rather than citing their own papers which would give a wrong impression.”

Reply to remark 8:

The reviewer is definitely right, our choice of references was indeed inadequate. This has been addressed by making reference to papers by Dellnitz and Junge 1999, Junge and Koltai 2009, and Froyland 2007. The previous reference has been put in the context of applications on climate model data.

Referee 2.

Remark 1: “Page 16/17 – Figure 9: I had difficulties to understand why the authors chose a σ value (12%) that makes the detection of the cold-state (C-state) impossible to view (see Fig. 9), while lower σ -values have been used to construct Fig. 8. I think one or two panels could be added to Figure 9, which also show the C-state, maybe using a lower σ -value or a shorter lifetime. If this is not feasible, I would ask the authors to more clearly explain that to the reader in section (iii) on page 16/17.”

Reply to remark 1:

This issue is now addressed at page 15 before figure 7. We have added few sentences as follows:

Here one faces a typical dilemma in terms of optimal use of computational resources. Considering a weaker noise would in principle facilitate the detection of the C state, and, in general, of the finer features of the phase space of the system. On the other hand, the exploration of the phase space of the system becomes more difficult, as the stochastic orbit is trapped for a very long time near either the W or the SB state, and the visits to the C state (unless the initial condition is set very close to it, as done below) are extremely unlikely. Hence, it is hard to obtain a reasonably good estimate of the invariant measure given finite computational resources. Note also that, as discussed in [50], in the zero noise limit the invariant measure concentrates on the state featuring the lowest value of the quasi-potential (the SB state, in this case). As discussed below in Sect. 4(c), harnessing methods of data

science, and specifically manifold learning, allow us to sort out such a conundrum and to automatically detect with high statistical significance also the C state in the case $\sigma = 12\%$.

At page 16 we also explain more carefully that we perform simulations targeted at describing escape processes from the C state, where we need the knowledge of the deterministic C state.

Remark 2: “Page 19, line 39-41: The authors write “However, the distribution of the air temperature variables in this state differs significantly from the C state in setup B (not shown)”. Although, it is clearly disentangled what the detected green state is and what its difference to the C state is, I would feel it would be better to show the air temperature of the green state and the C state in a supplementary Figure (instead of writing “not shown”).”

Reply to remark 2:

We thank the reviewer for having suggested us to look into this. We have added a section (6) and a figure (9) in the supplementary material. We show that the spurious state is colder than the C state but indeed, there is some correspondence between the meridional structure of the temperature and wind fields. We interpret this as the fact that the spurious state is a dynamical remnant of the C state (see reference 119 discussing the so-called ghost states).

Minor comments (including some typos):

1) Abstract: The authors write “... in one of the two climate models ...”, but actually only one climate model (PLASIM) is used in two different setups. This should be clarified. Also, the name of the model could be named in the abstract

Done.

2) Page 2, line 28: The authors could consider citing another EMIC apart from the CLIMBER-2/3alpha family (e.g. Bern3D, Loveclim, ...)

Done.

3) Page 3, line 11: A full stop is missing after source “..., 49)”

Done.

4) Page 3, line 37: A comma should replace the point after source “...,65).”

Done.

5) Page 4, line 28: “asymptotic states”  “asymptotic state”

Done.

6) Page 8, line 28/29: The authors state that PLASIM cannot represent the process of deep water formation. Do the authors have an intuition of how the results might change in case the model could do that?"

We thank the reviewer for this comment. The following sentences have been added in the first paragraph of Section 4

The lack of a realistic dynamic ocean implies that PLASIM misses the multidecadal, ultra-low frequency relaxation and oscillatory modes associated with the advective feedbacks of the ocean near the W climate, see discussion in [98]. Additionally, as discussed later in Sect. 5, the presence of a dynamic ocean might lead to the presence of additional features in the quasi-potential defining the W climate, associated with the presence of tipping points. Instead, one expects that the lack of a dynamic ocean is less critical near the SB climate, because no large-scale circulation is present when the surface of the Earth is entirely frozen.